

# Modelling biocide and herbicide concentrations in catchments of the Rhine basin

Andreas Moser[1], Devon Wemyss[1#], Ruth Scheidegger[1], Fabrizio Fenicia[1], Mark Honti[2], Christian Stamm[1]

[1]Eawag – Swiss Federal Institute of Aquatic Science and Technology, Dübendorf, 8600, Switzerland
[2]MTA-BME Water Research Group, Hungarian Academy of Sciences, Budapest, 1111, Hungary
[#]current address: ZHAW School of Management and Law, Winterthur, 8400, Switzerland

*Correspondence to*: Christian Stamm (christian.stamm@eawag.ch)

**Abstract.** Impairment of water quality by organic micropollutants such as pesticides, pharmaceuticals or household chemicals is a problem in many catchments worldwide. These chemicals originate from different urban and agricultural usages and are transferred to surface waters from point or diffuse sources by a number of transport pathways. The quantification of this form of pollution in streams is challenging and especially demanding for diffuse pollution due to the high spatio-temporal concentration dynamics, which requires large sampling and analytical efforts to obtain representative data on the actual water quality.

Models can also be used to predict information to which degree streams are affected by these pollutants. However, spatially distributed modeling of water quality is challenging for a number of reasons. Key issues are the lack of such models that incorporate both urban and agricultural sources of organic micropollutants, the large number of parameters to be estimated for many available water quality models, and the difficulty to transfer parameter estimates from calibration sites to areas where predictions are needed.

To overcome these difficulties, we used the parsimonious *iWaQa* model that simulates herbicide transport from agricultural fields and diffuse biocide losses from urban areas (mainly façades and roof materials) and tested its predictive capabilities in the Rhine River basin. The model only requires between one and eight global model parameters per compound that need to be calibrated. Most of the data requirements relate to spatially distributed land use and comprehensive time series of precipitation, air temperature and spatial data on discharge.

The model was calibrated with data sets from three different small catchments (0.5 – 24.6 km$^2$) for three agricultural herbicides (isoproturon, S-metolachlor, terbuthylazine) and two urban biocides (carbendazim, diuron). Subsequently, it was validated for different years on 12 catchments of much larger size (31 – 160'000 km$^2$) without any modification. For most compound-catchment combinations, the model predictions revealed a satisfactory correlation (median r$^2$: 0.5) with the observations and the peak concentrations mostly predicted within a factor of two to four (median: 2.1 fold difference for herbicides and 3.2 for biocides respectively). The seasonality of the peak concentration was also well simulated, the predictions of the actual timing of peak concentrations however, was generally poor.

Limited spatio-temporal data, first on the use of the selected pesticides and second on their concentrations in the river network, restrict the possibilities to scrutinise model performance. Nevertheless, the results strongly suggest that input data and model structure are major sources of predictive uncertainty. The latter is for example seen in background concentrations that are systematically overestimated in certain regions, which is most probably linked to the modelled coupling of background concentrations to land use intensity.





Despite these limitations the findings indicate that key drivers and processes are reasonably well approximated by the model and that such a simple model that includes land use as a proxy for compound use, weather data for the timing of herbicide applications and discharge or precipitation as drivers for transport is sufficient to predict timing and level of peak

concentrations within a factor of two to three in a spatially distributed manner at the scale of large river basins.

## 1 Introduction

Mankind uses thousands of synthetic chemicals for many different purposes in households, industries or agriculture (Schwarzenbach et al., 2006;Bernhardt et al., 2017). Many of these compounds reach water bodies during some stage of their life cycle. Accordingly, the impairment of water quality caused by substances such as pharmaceuticals, household chemicals

or pesticides is a problem of many catchments worldwide. From an ecological point of view, pesticides are often of special concern because they have been designed to harm a wide range of organisms.

Pesticides are used for different purposes. In agriculture they are used to protect crops from weeds, pests or diseases. However, the same compounds may be also used to fight unwanted organisms on materials such as roofs, façades or ships.

Depending on where pesticides are used, they may reach water bodies via different path ways. Although pesticides may be ecotoxicologically relevant chemicals even in treated waste water discharged from point sources (Munz et al., 2017;Müller et al., 2002) diffuse pollution is often dominant for these compounds (Moschet et al., 2014). The quantification of this form of pollution in streams is challenging due to the high spatio-temporal concentration dynamics, which requires large sampling and analytical efforts (e.g., Wittmer et al., 2010;Leu et al., 2004b).

As a consequence, the water quality status of many water bodies is not quantified sufficiently for properly addressing management and research questions that require a proper understanding about spatio-temporal patterns of pesticides occurring in streams. There may be deficits with regard to the spatial or temporal coverage of data as well as coverage of all chemicals of interest (Moschet et al., 2014).

Spatially (semi-)distributed models can potentially fill such gaps and have been developed and used for decades to do so (Borah and Bera, 2004). Some of these models (e.g. SWAT (Arnold et al., 2011), GREAT-ER or MONERIS (Berlekamp et al., 2007)) have been widely used, many others have been developed and used in specific research contexts (e.g, ZIN-AgriTra (Gassmann et al., 2013) One of the challenges related to modelling diffuse pesticide losses is the necessity to cover all relevant sources and flow paths. Many models for example, do not simulate urban and agricultural processes with the

same level of detail. This may pose a serious problem in regions that are characterized by a mixed land use of urban and agricultural areas such as in many parts of densely populated Central Europe.

Models differ widely in the degree to which they aim to represent explicitly the relevant processes. On the one hand, so-called physically based models try to describe them with equations in such a way that the model parameters should have a

real physical, chemical, or biological meaning independent of the model application with the goal to provide causal system understanding (Bossel, 1994;Beck, 1987). Generally, running such highly parameterized models comes with a huge data demand, and – as this demand usually cannot be covered – many model parameters cannot be estimated from independent observations. In the end, this leads to either the use of potentially unrealistic parameter values or calibration, the latter facing the problem that many of the parameter values cannot be properly identified possibly inducing large uncertainties during a

validation or prediction phase (Beck, 1987;Brun et al., 2001).



On the other hand, more conceptual, parsimonious models try to cope with the lack of (spatially distributed) data by dramatically reducing the number of parameters. This comes at the cost that model parameters may lose their direct physical or chemical interpretation. Such parsimonious models basically assume that essential aspects of the response of a complex (real) system can be represented by some rather simple mathematical descriptions that incorporate the effects of major

external drivers, such as precipitation. Such types of models are frequently used in hydrology for simulating discharge (e.g., Beven and Kirkby, 1979), there are also applications to water quality simulations (Hahn et al., 2013;Jackson-Blake et al., 2017) but only few models for simulating pesticide transport to surface waters (Honti et al., 2017).

Here we present a model that covers major urban and agricultural sources for pesticides in streams that can be applied to
large water basins, provides high spatial and temporal resolution and is still parsimonious. It is similar to the *iWaQa* model approach in (see e.g., Honti et al., 2017) but adapted for large basins by including an explicit routing component. It differs from many other model concepts in that it does not include a rainfall-runoff module but links agricultural pesticide losses to measured discharge and urban biocide losses directly to precipitation.

Specifically, the paper has the following objectives:

1. Description of the model concepts and their implementation.
2. Calibration of the model on selected small catchments and selected pesticides representing agricultural herbicides and urban biocides
3. Evaluating the performance of the calibrated model with blind predictions on a large set of validation catchments. This step includes a pronounced spatial upscaling of the model by three to four orders of magnitude.

We have used the Rhine basin as a case study to investigate these questions.

## 2 Study area

The study is carried out in the Rhine basin upstream of the gauging station Emmerich am Rhein (Germany; see Fig. 1). We limited the analysis to this part of the basin because the model structure does not cover complex, strongly managed flow regimes as prevalent in the Dutch part of the basin. Even with these restrictions, the study area is one of the largest drainage basins in Europe with an area of 160'000 $km^2$ covering land of eight countries, mainly from Switzerland, Germany, France and Luxembourg. The total length of the river network is 63'080 km and is divided into more than 30'000 catchments
according to the CCM River and Catchment database for Europe, version 2 (CCM2) from Vogt et al. (2007).

Altitude ranges from above 4200 m.a.s.l. in the Bernese Alps in the south to 17 m.a.s.l. at Emmerich in the north. Accordingly, the hydrological regime varies strongly across the basin. Discharge regime in the southern part of the basin is largely influenced by snow accumulation and melt. As a consequence, most southern rivers are of pluvio-nival type with low water periods during winter and flood events occurring mainly in summer. In contrast, sub-basins further north (Neckar,
Main, Moselle, Ruhr, etc) are characterized by a pluvial regime with winter floods and low water levels in summer. Similarly, temperature regimes show important differences, which may be reflected in shifts in phenology of crops and hence in application periods of agricultural pesticides.

The basin is densely populated (290 inhabitants $km^{-2}$ in the study area) with strong regional differences. Arable cropping is
an important land use in large parts of the basin. More details on specific crops and their spatial distribution are presented in the Supplementary Material (Fig. S3).





The Rhine River is heavily used by hydropower plants upstream of Iffezheim along the main channel and main tributaries. However, the effects on travel times is rather moderate on the streams of interest and was neglected during the routing calculations.

## 3 Model description

### 3.1 Spatial discretization and model structure

The river basin is discretized into subcatchments based on the CCM2 database. To reduce the number of subcatchments and ensure a reasonable minimum size, CCM2 catchments smaller than 2 km$^2$ were merged with the next downstream catchment.

The resulting 18'240 subcatchments with an average area of 8.8 km$^2$ are the primary computational units of the model. Further details on the spatial representation are provided in Appendix A2 of the Supplementary Material.

The model consists of two principal components. The first component – the *substance transfer module* – simulates the transfer of the pesticides from their point of use (e.g., the fields to which herbicides are applied) to the outlet of each subcatchment. The second component – the *routing module* – links the contribution of all subcatchments, and represents the

in-stream transport and fate processes of the chemicals.

We assume that subcatchments are laterally disconnected from each other, and therefore simulations of the substance transfer module can be run separately for each subcatchment. Subsequently, the routing module integrates all outputs of the substance transfer module by processing subcatchments from up- to downstream.

For the routing, the main river (and optionally also tributaries) is split into river segments (see Appendix A2, Fig. S2). Each

segment receives input from upstream and lateral directions as well.

### 3.2 Substance transfer module

This module consists of several independent parts that describe the transfer of chemicals from the different pesticide sources in the catchment. In particular, it consists of the *iWaQa* model describing substance transfer for herbicides (Section 3.2.1)

and another for biocides (Section 3.2.2). These models treat subcatchments as spatially lumped units. The models are very parsimonious such that they only require one to eight empirical, yet global model parameters per simulated chemical (Table 1). All other model inputs consist of (generally) available statistical data on chemical consumption, spatial data on land use and hydro-climatic time-series.

.

### 3.2.1 Substance transfer for herbicides

This section describes first the system of the herbicide model and subsequently the input and output of the system.

This model consists of two spatially lumped storage terms representing the dissolved and sorbed fractions of the total herbicide mass $M(t)$ [g] in the topsoil layer of agricultural fields in the subcatchment. The first storage is the mass dissolved in the pore water $M_w(t)$ [g] being instantly available for release to the river. The other represents mass adsorbed to the soil

matrix $M_s(t)$ [g] and is unavailable for immediate release.





The exchange between the two storages is described by two kinetic rate parameters: sorption to the soil matrix is described with the transfer rate $k_{w-s}$ [d⁻¹] and the reverse flux with $k_{s-w}$ [d⁻¹], respectively. Both stocks degrade according to first-order kinetics with decay rate $k_{deg}$ [d⁻¹].

The mass balance and the two first-order differential equation describing the change in stock of herbicide masses in the system are given by:

$$M(t) = M_w(t) + M_s(t) \tag{1}$$

$$\frac{dM_w}{dt} = \rho \cdot \dot{M}_a(t) - k_{w-s} \cdot M_w(t) + k_{s-w} \cdot M_s(t) - k_d \cdot M_w - L_{hebicide} \tag{2}$$

$$\frac{dM_s}{dt} = (1 - \rho) \cdot \dot{M}_a(t) - k_{s-w} \cdot M_s(t) + k_{w-s} \cdot M_w(t) - k_d \cdot M_s \tag{3}$$

where $\dot{M}_a(t)$ [g d⁻¹] the mass rate applied in the catchment during the application period and $\rho$ [–] represent the fraction of the applied mass that is immediately available for transport. The output $L_{herbicide}$ [g d⁻¹] is the herbicide load released from the current application at the outlet of the subcatchment.

*Input:*

Crop development and hence also the timing of herbicide applications is strongly controlled by temperature conditions in any particular year. As application dates are generally unknown, a temperature sum models is used to simulate crop growth and the related herbicide applications, which is linked to specific growth stages of the crops. In particular, we assume that application of herbicides starts when the daily temperature sum at a given location reaches a crop-specific temperature threshold (Honti et al., 2017). Daily mean values of temperature are summed up ($T_{sum}(t)$), though a restart is forced after freezing days. Once the objective temperature $T_{obj}$ is reached, 1/14 of the total application mass is applied on each following rain-free day until the total application mass is depleted. Herewith the selection of a universal application date is avoided and the method accounts for regional climatic differences.

*Output:*

The concept to describe the transfer of the applied herbicides from the fields to the river is based on the empirical observation that herbicide concentrations increase with flow during discharge events during the application period (Leu et al., 2010). Mechanistically this can be explained by the occurrence of fast transport processes (with high herbicide concentrations) such as surface runoff and fast subsurface flow through drainage systems or macropores (Leu et al., 2004a) during discharge events. Hence the concentration (C [g m⁻³]) in the river is described – in a first approximation – as proportional to the discharge $Q(t)$ [m³ d⁻¹] in the case of a recent application on the fields; the load [g d⁻¹] increases quadratically with discharge:

$$C_{herbicide}(t) = \alpha(t) \cdot Q(t) \tag{4}$$

$$L_{herbicide}(t) = C(t) \cdot Q(t) = \alpha(t) \cdot Q(t)^2 \tag{5}$$

where $\alpha$ [g d m⁻⁶] is the proportionality coefficient relating the magnitude of the discharge to the released loads.

The proportionality coefficient depends on $M_w(t)$, the mass dissolved in the pore water and instantly available for release:

$$\alpha(t) = \varepsilon \cdot M_w(t) \tag{6}$$



where ε [d m$^{-6}$] becomes a catchment-independent, empirical loss factor that needs calibration for each chemical (see sec. 4.2).

Certain herbicides are present in significant concentration outside of the application period too (see for example (Leu et al., 2004a)). Therefore, we added a constant background concentration ($C_{back}$ [g m$^{-3}$]) to the substance transfer model. This step was essential to ensure a proper calibration of the model. By doing so we implicitly assume a constant concentration of herbicides independent of the application period, representing e.g. other, not seasonal sources or a general presence in the baseflow due to the long-term persistence of pesticides in groundwater. Thus, the total released load of the system becomes:

$$L_{release}(t) = C_{back}(t) \cdot Q(t) + \varepsilon \cdot M_w(t) \cdot Q(t)^2 \qquad (7)$$

### 3.2.2 Substance transfer for biocides

Biocides are applied in the urban settlement on façades, flat roofs, basement seals and underground parking lots. Due to the potential year-round application and the long-term protection purpose of biocides, it is assumed that the stock in the urban

settlement is constant over time (Wittmer et al., 2010).
The leaching of biocides in urban areas is a complex process and several studies provide quantitative information on loss rates, dynamics and driving factors (Jungnickel et al., 2008;Burkhardt et al., 2008;Wittmer et al., 2011). The process is mainly driven by precipitation when water flows over the treated surfaces and it was observed that concentration patterns of urban compounds follow the rainfall pattern more than the river discharge (Wittmer et al., 2010). Therefore, the current

model simplifies the processes by assuming the release being proportional to precipitation and instantaneous transport to the rivers. The following equation thus describes the resulting modelled load $L_{biocide}(t)$ [g d$^{-1}$] to the rivers:

$$L_{biocide}(t) = M_a \cdot \beta \cdot P(t) \qquad (8)$$

With $M_a$ [g] the total mass present in the catchment within the model period, $\beta$ [m$^{-1}$] the substance-specific loss rate (to be calibrated, see below), $P(t)$ the precipitation [m d$^{-1}$].

### 3.2 Routing module

*Load aggregation*

Concentrations of micropollutants at the outlet of any catchment composed of several subcatchments are predicted by aggregating the loads from the output of the substance transfer module and division by the actual total discharge. The approach considers the local availability of sources and the spatial distinctions of the driving factors (discharge or precipitation). However, instantaneous aggregation assumes no in-stream losses, such as degradation, sedimentation or diffusion taking place during the transport. Furthermore, it implies that the temporal resolution should be larger than the

longest travel time of a component during a rain or discharge event. Otherwise the concentration dynamics are affected.
A special situation is given by the presence of the large pre-alpine lakes (Lake Constance, Lake Lucerne etc.) in the river network. Because of the long water residence time in these water bodies (months to years), the concentration dynamics in the lake outlet are strongly dampened and differ substantially from other river sections. To account for these different dynamics, we simulated the input into each of these lakes separately by the substance transfer module. We assumed complete mixing



into one year of discharge and used the resulting concentration as a constant value in the river water flow out of the respective lakes. The load varied accordingly with discharge from the lake. A different case was Lake Biel, which was not treated as a mixing reactor because of the short spatial distance between the inflow and the outflow of the river Aare.

*Routing with AQUASIM*

In larger river basins the effects of travel time, dispersion and degradation during pollutant transport in the river system become more important. The assumption of instant arrival of pollutants at the outlet within daily time steps does not hold true anymore and hydraulic routing becomes indispensable.

To that end the load output from the substance transfer module was used as input into the program AQUASIM (Reichert, 1994) that was used for describing the transport and fate processes within the main rivers. Flow was described with the

kinematic wave approximation of the St. Venant equations. Transformation and sedimentation through sorption was neglected because the model compounds are sufficiently stable and show only weak sorption (see also Honti et al. (in preparation)). .

# 4 Methods

## 4.1 Model input data

### 4.1.1 Discharge, precipitation and temperature

Hourly discharge data was obtained for 1033 stations from federal and national agencies (see Appendix A4, Table S4) to derive two kinds of discharge time series for all subcatchments. The first, termed local runoff, refers to surface and subsurface runoff originating from the specific subcatchment and is used in the substance transfer module for herbicides. The

other is the streamflow at the outlet of a subcatchment required in the routing module to calculate the concentrations of catchments or as input to AQUASIM. For headwater subcatchments without any further upstream connections, the local runoff is identical to the streamflow.

Time series of local runoff are derived from the records of gauging stations measuring rivers with a Strahler stream order

(Strahler, 1957) less than five. Using gauging stations at larger rivers would not accurately reproduce the high temporal variations of the local runoff. The recorded discharge is allocated to the subcatchments upstream according to the drainage area ratio method (Hirsch, 1979). Unfortunately, many subcatchments remain ungauged hereby. On one hand this method does not provide time series for subcatchments downstream of the stream gauges with Strahler order larger than four, on the other hand numerous ungauged tributaries join the river network downstream of the selected stream gauges. In both cases a

nearby reference stations (with Strahler order < 5) is selected and the area ratio method is applied to calculate local runoff. Selection of the reference stations is based on the map-correlation method from Archfield and Vogel. (2010) calculating the correlation between stream gauges and choosing the station with the most correlated discharge at the considered catchment.

The stream flow time series for all subcatchments were deduced in a similar way. Upstream of stream gauges with Strahler

less than five, the discharge is allocated according to the drainage area ratio and accumulated towards downstream. The discharge of any stream gauge is passed on to the downstream subcatchments and accumulated with the streamflow of converging tributaries. Likewise to the local runoff, the streamflow for ungauged tributaries is adapted from reference stations selected with the map-correlation method.





Hourly precipitation data for the study area is available for Switzerland from MeteoSwiss CombiPrecip (Sideris et al., 2014) and for the rest of the Rhine basin from RADOLAN (Bartels et al., 2004), a product of the German Meteorological Office (DWD). Both are raster datasets (with a spatial resolution of 1 km$^2$) computed using geostatistical combination of radar sensing and rain gauge measurements. Small temporal gaps in the precipitation data or uncovered parts in the French region

5 were filled with data from the nearest available rain gauge. Additional data from rain gauges is available for Luxembourg and France. By intersecting the raster cells with the subcatchments, the most accurate conversion was achieved with the area-weighted mean of the overlapping grid cells within a subcatchment.

Raster temperature data with daily mean values are retrieved from MeteoSwiss TabsD (Begert et al., 2003) with a spatial

10 resolution of 0.02 deg (~2.3x1.6km) and from the European dataset termed E-OBS (Haylock et al., 2008) with a coarser resolution of 0.25 deg (~27.8x18.8km). Both datasets are spatial interpolations of monitoring stations.
Given that the Swiss temperature dataset has a finer grid size than the average area of the subcatchments (8.8 km$^2$), it allowed for estimating reliable mean temperatures for all subcatchments in Switzerland. The grid size of the E-OBS temperature data was significantly larger than the average subcatchments. The spatial resolution of the E-OBS temperature

15 data set was therefore refined using a Digital Elevation Model with a grid size of 1 km$^2$ (the DEM was obtained from the GMES RDA project, EEA, 2013). In particular, the deviation between the altitude of the DEM cells and the E-OBS cells was calculated. These deviations were multiplied with a temperature lapse rate of -6.5 °C/km and added to the temperature values of the E-OBS cells. Thus a gridded temperature model with a resolution of 1 km$^2$ was obtained.

### 4.1.2 Land use data

20 Herbicides are applied on specific crops, therefore detailed, spatially distributed agricultural land use data were required. The dataset "Agricultural Landuse2000" from the JRC AFOLU project (Leip et al., 2007) classifies agricultural land use into 30 crops and for a grid with a resolution of 1 km$^2$ by combining remote sensing with statistical information of the agricultural production.
This European dataset on agricultural land use does not cover Switzerland. In order to have a dataset with the same crop

25 categories and a similar spatial resolution, a harmonized dataset was created from the Land Use Statistics of Switzerland (Swiss Federal Statistical Office FSO, 2012) and the census of agricultural enterprises (Swiss Federal Statistical Office FSO, 2011). The cultivation areas of 60 listed crops reported in each municipality in the census were distributed on the grid cells of the Land Use Statistics belonging to the 3 agricultural land use classes, leading to an average fraction of cultivated area of crop $l$ per grid cell in community k:

$$W_k^{(l)} = \frac{a_k^{(l)}}{G_k^{(tot)}} \tag{9}$$

$W_k^{(l)}$ [–] is the average fraction of crop $l$ being cultivated in a single grid cell belonging to community $k$. The $a_k^{(l)}$ [ha] is the cultivation area of crop $l$ (reported in the census) in municipality $k$, $G_k^{(tot)}$ [ha] is the sum of the area of all agricultural grid

35 cells in community $k$. The 60 crop categories of the census are merged to the 30 categories from the European "Agricultural Landuse2000", thus a consistent database is accomplished with a comparable approach of distributing statistically reported areas to spatial land use data.

Land cover of housing and settlements is available with vector based maps, where every building is precisely represented by

40 a polygon and in some cases with knowledge about its height.





- France:              Institute géographique nationale (IGN) BD TOPO® (with height)
- Germany:         Arbeitsgemeinschaft der Vermessungsverwaltungen Deutschland (ADV) ALKIS®
- Luxembourg:   Administration du Cadastre et de la Topographie (ACT) BD-L-TC
- Switzerland:    Federal Office of Topography (swisstopo) swissTLM 3D (with height)

Façade surfaces are calculated by multiplying the contours of buildings with their height where available (CH, FR). For the other countries (DE, LU) the façade areas within a subcatchment are estimated from the footprints areas and the population. Footprint and façade follow a linear relation, whereas the relationship between population $N_{pop}$ [–] and façade $A_{fac}$ [m$^2$] appear to be polynomial . With the Swiss data the following regression was obtained:

$$A_{fac} = 1.55 \cdot A_{foot} + 1.45 \cdot 10^5 \cdot N_{pop} + 6.20 \cdot 10^{-4} \cdot \left(N_{pop}\right)^{0.49} \tag{10}$$

This regression was validated with the French data achieving reasonable results and finally used to calculate the façade areas in Germany and Luxembourg (see Supplementary Information, Appendix A6, Fig. S5).

**4.1.3 Model compounds, use and sale data**

Five model compounds (see Tab. S1) have been selected for this study: three agricultural herbicides (isoproturon (IPU), S-metolachlor (MEC), terbuthylazine (TBA)) and two (dual use) biocide (carbendazim (CBZ), diuron (DIU)). The biocides are mainly used in urban environments to protect materials. They may also have some agricultural use in some regions of the basin (e.g., in Switzerland) but the usage is of minor relevance and is neglected here.

Use and consumption data for the chemicals are not available in a spatially distributed manner. To provide input for all spatial model units, we proceeded in two steps. First, we obtained statistical data on use/consumption data for regions or countries. Subsequently, we downscaled these statistical data based on land use or population.

Annual sales data of herbicides were available from the countries Switzerland (Agroscope ZA-AUI, (Spycher and Daniel,
2013)), Germany (Federal Office of Consumer Protection and Food Safety, (Federal Office of Consumer Protection & Food Safety BVL 2008 - 2012)) and the French regions Alsace (Office national de l'eau et des milieux aquatique, (Office national de l'eau et des milieux aquatique ONEMA, 2014)) and Lorraine (Groupe Régional d'Action contre la Pollution Phytosanitaires des Eaux Lorraine, (Groupe Régional d'Action contre la Pollution Phytosanitaires des Eaux Lorraine GRAPPE Lorraine, 2005)) for the years 2008-2012 (except the study for Lorraine was only issued for 2005). The spatial
coverage area of the statistics varied strongly ranging from 357'300 km$^2$ for Germany to 8'330 km$^2$ for Alsace. The Swiss dataset only provided coarse ranges of substance sold per year from which the mean values were used.
Only one source for the use and sale of biocides was at hand. The survey of Burkhardt & Dietschwiler (2013) investigated the consumption rates in Switzerland of various biocides in antifouling paints, masonry and wood protection agents. The use rates have been applied to the entire study area.

The mass distributed on the agricultural fields respectively applied on houses of each catchment was estimated by downscaling regional or national sales data $\dot{M}_{tot}$ [g d$^{-1}$] with the ratio of the local application area (area within a subcatchment) $A_a$ [ha] to the total application area $A_{tot}$ [ha] (total area within the considered sales study):

$$\dot{M}_a = \frac{A_a}{A_{tot}} \dot{M}_{tot} \tag{11}$$





The application area was distinct for use classes and substances. For herbicides it was the sum of possibly treated agricultural land use areas, more specifically the crops for which a substance is authorized and primarily used. Biocides were applied on façades of a building. The sum of the respective building surface composes the application area.

## 4.2 Calibration of the catchment model

### 4.2.1 Calibration sites

To calibrate the model, data from field studies was used that provided simultaneously data on application amounts of substances as well as on losses to the rivers. Such studies are rare and we used the following studies situated in the northeastern part of Switzerland. The sampling campaigns from Gomides Freitas et al. (2008) and Doppler et al. (2012) measured herbicide concentrations at the small-scale agricultural catchments Summerau and Ossingen, respectively, after a controlled herbicide application. Wittmer et al. (2010) monitored the mass and dates of herbicide applications in a slightly larger catchment Mönchaltorf (25 km$^2$) with mixed land use. The biocide application was estimated with product and statistical information. Subsequently the losses from the catchments were measured at the outlet of the catchment.

### 4.2.2 Calibration procedure

The substance-specific parameter sets for herbicides $\theta_{herbicide} = \{\varepsilon, C_{back}, \rho, k_{w-s}, k_{s-w}, k_{deg}, T_{obj}\}$ and for biocides $\theta_{biocide} = \{\beta\}$ cannot be measured and require calibration. Parameter $T_{obj}$ regulating the timing of herbicide application was only calibrated in the case of Mönchaltorf where regular application occurred at the farmers' chosen timing. At Ossingen and Summerau the application was experimentally controlled and therefore a calibration of $T_{obj}$ would be meaningless.

The model parameters were calibrated using a Bayesian inference approach. The likelihood function accounted for the parameter uncertainty and the structural model errors. For herbicides model errors were assumed to deviate stronger during the application season. Therefore an error-scaling function was added depending on the substance input to the system and a driver imitating the approximate substance application to the fields. The additional parameters to calibrate resulting from the error-scaling function were $\theta_{herbicide,error} = \{\mu, \sigma_{error}\}$ where μ is a scaling factor for the substance input and $\sigma_{error}$ the calibrated standard deviation of the total model error. For the biocides the error variance was assumed to have no seasonality.

Measured peak concentrations of herbicides in the calibration studies occurring before the monitored application period were excluded from the calibration procedure as they represent accidental spills or runoff from hard surfaces. As such events are not represented in the model, including them would have spoiled the identification of model parameters.

The likelihood function used in this study is based on the assumption that Box-Cox transformed time series of concentration data $C$ lead to independent and identically distributed normal errors. The likelihood function is as follows:

$$p(C_{obs}|\theta) = \left(\frac{1}{\sqrt{2\pi\sigma^2}}\right)^N exp\left(-\frac{1}{2\sigma^2}\sum_{i=1}^{N}\left(g\left(C_{obs,i}\right) - g\left(C_{mod,i}\right)\right)^2\right)\prod_{i=1}^{N}\frac{dg(C_{obs,i})}{dC} \tag{13}$$

where $\sigma^2$ is the error variance, $N$ is the total number of observations in all subcatchments, $C_{obs}$ and $C_{mod}$ are the observed and the modelled concentrations for the data point $i$. The transformation $g(\cdot)$ is the Box-Cox transformation used to remove the heteroscedasticity of the residuals:





$$g(C) = \frac{c^{\lambda}-1}{\lambda} \qquad (14)$$

The parameter $\lambda$ was set to 0.3.

The Jacobian of the transformation $\frac{dg(C_{obs})}{dC} = \prod_{i=1}^{n} C_{obs,i}^{(\lambda-1)}$ was required to compensate the distortion of the likelihood by using the transformed variables.

### 4.2.3 Prior distributions

Priors for the substance-specific loss rates were estimated based on reported information in the calibration studies (see Appendix A8, Table S4). Estimation for the substance-specific $\varepsilon$ of the herbicide model is based on the reported loss rates from the studies. Neglecting background concentrations the time-averaged concentration $\bar{C}$ during the main loss period from $t_0$ to $t_{0end}$ is given according to Eq. 4 as

$$\overline{C_{herbicide}} = \frac{\varepsilon}{(t_{end}-t_0)} \times \int_{t_0}^{t_{end}} M_w(t) \times Q(t)dt: \qquad (15)$$

Based on measurements, $\bar{C}$ can also be expressed as:

$$\overline{C_{herbicide}} = \frac{\dot{M}_a \times \Delta\tau \times lr_{study}}{\int_{t_0}^{t_{end}} Q(t)dt} \qquad (16)$$

where $\dot{M}_a$ is the average application rate in the catchment, $\Delta\tau$ is the duration of the application period, $lr_{study}$ is the empirically observed loss rate from the study . From Eq. 14 and 15, it follows that $\varepsilon$ can be approximating as:

$$\varepsilon = \frac{\dot{M}_a \times \Delta\tau \times lr_{study}}{\int_{t_0}^{t_{end}} Q(t)dt \times \int_{t_0}^{t_{end}} M_w(t) \times Q(t)dt} \approx \frac{\dot{M}_a \times \Delta\tau \times lr_{study}}{(t_{end}-t_0) \times \overline{M_w} \times (\bar{Q})^2} \qquad (17)$$

where $\bar{Q}$ is the mean discharge during this period, and $\overline{M_w}$ is the mean calculated using the known application pattern and a first-order approximation for the sorption and decay.

Priors for the substance-specific loss rates of the biocide model was the total loss rate reported in Wittmer et al. (2010) divided by the yearly sum of precipitation. Having multiple study catchments or ranges of loss rates allowed to calculate a distribution of the priors for $\varepsilon$ and β.

Prior distributions for the parameters describing pesticide fate in the soil ($\rho$, $k_d$, $k_{w-s}$, $k_{s-w}$, $k_{deg}$) were derived from field experiments. The equations are fitted to the Freundlich adsorption isotherms with time-varying sorption coefficients measured in soil samples (Freitas et al. 2008).

The maximum of the posterior parameter distribution was found by performing a Nelder-Mead simplex optimization. The maximum likelihood parameter set was used as a prior for the Markov chain Monte Carlo (MCMC) simulation using the Metropolis algorithm (Gamerman, 1997). The developed posterior parameter distributions were used to predict the parameter and model uncertainty. The procedure was repeated for every calibration site separately.





### 4.3 Model validation and routing

Several comprehensive sampling campaigns from the Swiss "National Surface Water Quality Monitoring Program - NAWA" (Federal Office for the Environment FOEN, 2013) and data from a continuous monitoring station were selected to evaluate the model.

The first campaign (called SPEZ 2012) comprised five catchments (Fig. 2) ranging from 39 to 105 km$^2$ and have varying extents of urban and agricultural influences (Appendix A7, Table S3). The measurement campaign was accomplished from March to July 2012 with biweekly time-proportional mixed samples (Moschet et al., 2014).

The second survey was the "National River Monitoring And Survey" termed NADUF, where weekly or biweekly mixed-samples were taken during 2009 (Stamm et al., 2012). The monitoring sites were in the north-eastern part of Switzerland and

quantified the concentrations of several organic micropollutants in five nested catchments. These nested catchments have a large range of size from 74 km$^2$ to 14'718 km$^2$ comprising between 22 and 2554 subcatchments.

A third validation was conducted with data for 2011 from the continuous measurement program of the International Rhine Monitoring Station (IRMS) near Basel. With five probes distributed over the cross-section, daily discharge-proportional pollutant levels are evaluated. The upstream area of the Rhine at this point covers almost 36'000 km$^2$ including the sub-

basins Alpine Rhine, Lake Constance, High Rhine and Aare.

Modelled hourly concentrations were adapted to the sampling periods of the respective validation surveys. According to the aggregation periods of mixed samples in the measurement surveys, the modelled concentrations were averaged over the sampling time periods, such that the resulting time series were fully comparable.

The issue of routing arises for larger catchments where the transport time is longer and also the processes along the way become more significant. For the sites of the NADUF survey the concentrations at the outlets were first modelled with load aggregation and in a second step river segments were defined where the routing with AQUASIM was calculated. Thus the influence of a physically-based hydraulic routing can be compared to the situation where in-stream transport and processes

are neglected.

In the case of the IRMS, measuring a large sub-basin of the Rhine, the catchment model is applied for 5950 subcatchments. Downstream of the lakes the substance transport was modelled with AQUASIM for the larger rivers (Rhine, Aare; Appendix9, Fig. S6). The simplistic approach with load aggregation was applied on this large scale as well.

### 4.4 Model predictions within the Rhine basin

The calibrated model was finally applied to the Rhine and the major tributaries to characterize the pollutant dynamics of herbicides. These simulations were real predictions without any further adjustments of model parameters. Due to the lack of statistical input data of the use of biocides in France and Germany predictions for the Rhine basin were not possible for carbendazim and diuron.

### 4.5 Technical implementation

The *iWaQa* model is written in C++ and the outputs are time series of concentrations, parameter estimations, posterior parameter distribution from the MCMC or matrices with the concentration predictions with the posterior parameters. Within a Python framework, i) the input for the substance transfer module is generated, ii) the substance transfer module runs the *iWaQa* model for the entire Rhine basin and iii) the two routing options are executed (see Appendix A1). Data preparation and analysis is effectuated with the programming language R (R Core Team, 2017).



All modules are executable individually. Preprocessing succeeds within 30 minutes to sort the hourly input data for all 18'240 subcatchments of the Rhine basin on an Intel x86 8-core processor. The substance transfer module takes approximately an hour to run and sort the output by both, subcatchments and time steps. Run times of the routing options differ substantially depending on the size of the considered catchment and the parameterization of AQUASIM. Generally the

load aggregation is calculated within a few minutes and the simulation of the main tributaries of the Rhine basin with AQUASIM is completed within 6 hours.

### 4.6 Model evaluation

Besides the likelihood used for parameter calibration, there are many metrics for evaluating model performance of hydrological and water quality models (Jachner et al., 2007;Smith and Rose, 1995;Reusser et al., 2009;Moriasi et al., 2007).

Out of those, we have selected some frequently used statistics (Table 3) that allow for a comparison with other studies. In addition, we have included some metrics that are more specifically designed to analyse aspects, which are of special relevance for this work. These measures include the Geometric Reliability Index of the cumulative distribution of the simulated concentrations to see how well the overall concentration level is met or the fold difference between the observed and simulated maximum concentration during the simulation period.

## 5 Results

### 5.1 Calibration

The calibration was carried out for all catchment-compound combinations for which observations are available (see Table 2). For the agricultural herbicides this provides several alternative calibration sets. The final decision of which set to use for further predictions was based on the performance in the validation step with the NAWA SPEZ sites (see below).

For the agricultural herbicides, the calibration resulted in a reasonable simulation of the observed concentration dynamics (Figure 3, Supplementary Material Fig. S7, S8). The calibrated uncertainty bands also followed the expected seasonal patterns: they were large during the application periods and decreased with time. The model, however, poorly captured the exact timing of the concentrations as one can see from the low Nash-Sutcliffe (NSE) coefficients (ranging between -0.05 and 0.62, median = 0.38; see Appendix A13, Table S10). Despite these deviations, the correlations between observations and

simulations were reasonable (range between 0.30 and 0.85, median = 0.68).

For the biocides, the model predicted a rather uniform distribution of concentration peaks around the year reflecting the precipitation patterns. The observations however, suggest a bi-modal seasonal pattern with higher concentrations in spring and fall. This pattern resulted in low correlations (r of 0.30 and 0.37; see Appendix A13, Table S10) and poor NSE values (-0.05 and 0.08).

The residuals pointed to systematic deviations between observed and modelled concentrations (Figure 4). The data group into two clusters. One of the clusters showed systematic underestimations of the observations, while the other showed the opposite. Comparison with the time-series revealed on the one hand that for most compounds, the highest observed concentrations peaks were (substantially) underestimated during calibration (see for example metolachlor or terbuthylazin in

Figure 3). These peak concentrations were underestimated by 13% to 83% (Table 4). On the other hand, the second cluster of data points indicates that concentrations of some (smaller) events were overestimated. This pattern suggests that the model structure did not capture the full dynamic range of the pesticide concentrations.

Despite these limitations, the concentrations were reasonably well represented by the model. The Geometric Reliability Index GRI indicates that the predicted concentrations of the agricultural herbicides were within a range of 1.9 to 2.5 of the



observations (Figure 5). When being interested in how well the cumulative concentration distributions are simulated (ignoring the timing) these values range between 1.4 and 2.2. As can be seen from Figure 5, the performance for the biocides was considerably poorer but the cumulative distribution was also reproduced better that the concentration time series.

Based on the relative RMSE one can compare the calibration performance across sites. Mönchaltdorf and Summerau yielded

better calibrations for metolachlor and terbuthylazin than the Ossingen data set (Appendix A13, Table S8) The opposite was true for isoproturon. In the case of Ossingen, a long dry period followed after the isoproturon application resulting in very low concentrations without a pronounced peak related to the recent application. This last aspect points to the fact that single calibration data sets may represent special situations hampering the predictive power during normal conditions. The application of metolachlor and terbuthylazine in Ossingen for example, took place later just before an intensive precipitation

event. Through direct shortcuts, such as manholes of drainage systems and storm drains, the transfer to the river was accelerated and very high concentrations have been measured (Doppler et al., 2012).

So far, we have compared the observations to the deterministic model predictions. Comparing the observations to the simulations including the prediction uncertainties due to the estimated parameter uncertainty (of the deterministic model)

and the total predictive uncertainty accounting for input and model structure deficits reveals that the parameter uncertainty contributes only a small fraction. Taking into consideration all sources of uncertainty leads to uncertainty bands that include most of the observations as can be seen from the cumulative concentration distributions depicted in Appendix A12, Figures S23, S24.

All calibrated parameters of the deterministic model had priors based on physical reasoning or empirical data, hence the maximum likelihood values are not expected to deviate strongly. This held true for the decay rate, the loss rates ($\varepsilon$ and $\beta$), the background concentration and the objective temperature. The parameters describing the herbicide (de-)sorption processes (initial availability $\rho$, transfer rates $k_{s-w}$ and $k_{w-s}$) changed considerably. In general, the sorption coefficient values were higher and degradation rates smaller than in a priori estimate, meaning that the available mass for release was smaller but

more persistent.

### 5.2 Validation

The different calibrated parameter sets were used to predict the corresponding concentrations for the validations case studies. To that end, the model output having a daily resolution was aggregated to the time periods of the real sampling strategies at the respective sites. In contrast to the calibration procedure, the validation step included also the simulation of the compound

input. This included the estimation of the applied amounts and the timing of the applications.

For the agricultural herbicides, several calibration data sets were available. All of them were first tested on the NAWA SPEZ sites. Based on their performance, one set per compound was used for simulating the larger NADUF and IRMS sites. Based on the correlation coefficients and the NSE criterion the parameter sets calibrated at Mönchaltorf for the compounds

isoproturon and terbuthylazine and the parameter set from Summerau for metolachlor were used for the other catchments (see Appendix A13, Table S8).

At the IRMS, the validation of the model was partially restricted due to the low concentrations that often remained below the limit of quantification (LOQ) of 5 ng/l for metolachlor and terbuthylazine. Nevertheless, concentrations were high enough to

evaluate the model performance during the application period.



The quality of the predictions varied between compound use class and the validation catchments. The GRI values demonstrate that the agricultural herbicides were simulated better than were the biocides (Figure 5). The distribution of observed concentrations was better represented by the model as compared to the actual time series. Interestingly, the model performed better in the larger catchments (Figure 5) despite the fact that calibrations were up-scaled to areas that are between

four and 70'000 time larger than the calibration catchment (Table 2).

The quality of the predicted maximum concentrations changed from the calibration to the validation step. While the values were systematically underestimated during calibration, this pattern changed substantially for the validation. Depending on the site-compound combination, the maximum concentrations were either clearly under- or overestimated (Table 4).

Irrespective of the sign of the deviation, the fold difference between observed and simulated concentrations mostly ranged between one and four (Figure 6). However, there were a few cases of extreme deviations because of either the observation of a pronounced and very high peak or very low measured values hardly exceeding the observed background. Again, the model performed better for the herbicides where for 50% of the predictions (site-compound combinations) the maximum concentrations were predicted within a factor of 2.0 deviation from the observations. For the biocides, the value was larger

(> 3.0). We observed also clear compound-specific patterns such as systematic over-estimation of diuron concentrations (see e.g., Appendix A12, Figure S28).

As during the calibration step, the Nash-Sutcliffe values were low pointing again to the problem of properly simulating the exact timing of concentration peaks (Figure 7). This was very pronounced for the biocides. The correlation coefficients

provided a mixture picture. For some compounds such as diuron, the correlations coefficient range between 0.29 and 0.68 (median = 0.56) for the NAWA SPEZ and NADUF sites. For others such as carbendazim or isoproturon the correlation was very variable especially between the NAWA SPEZ sites (see Appendix A13, Table S10). At the station on the Rhine in Basel, the correlations varied between being non-existing to fairly strong (isoproturon: r = 0.84 assuming load aggregation across the Rhine basin).

*Effects of routing*

For the IRMS measuring site, we compared the performance of the simple load aggregation procedure and the explicit routing with AQUASIM. Differences between both approaches were moderate. The routing yielded better results because some of the pronounced concentrations peaks predicted by load aggregation were substantially smoothed. Therefore, the

maximum concentrations were overestimated to a lesser degree. The median difference between observed and simulated maximum concentrations with and without routing were 3.1 and 3.4-fold, respectively (averages: 2.6 and 4.8, respectively). The slightly better NSE values also suggest a better performance with an explicit routing. These results provided evidence that at the scale of such large basins of 30'000 km$^2$ and beyond the explicit routing makes a relevant difference for pesticides studied at a daily resolution.

**5.3 Predictions for the Rhine basin**

Based on the findings reported above on the effects on routing, we only report the findings based on AQUASIM for the predictions of the main tributaries (Aare, Neckar, Main and Moselle) and the further measuring sites downstream of Basel. The total river length for which the routing was explicitly simulated with this module was 1773 km. We focus here on the three herbicides (isoproturon, metolachlor and terbuthylazine) because for them a minimum set of observations was

available.



The observed isoproturon concentrations revealed the two peaks in spring and fall as measured also at the other locations (Figure ). The model predicted the timing of the spring peak very well. Also the absolute concentrations level of the peak was simulated well (within 30% of the observation). Concentrations during the summer months were slightly underestimated; the fall peak was missed because no application was included in the model (see above).

The comparison of the simulated chemographs along the Rhine show some slight temporal shifts of the peaks caused by different application periods. Despite of the size of the basin however, these shifts due to varying phenology were small corresponding to a few days only.

The simulations show very similar patterns for the other two herbicides in the different tributaries (see Appendix A11, Figure S22). The time shifts between the different sub-basins were also very small. Unfortunately, these findings cannot be

tested against observations because the LOQ (10 ng/L and 50 ng/L for metolachlor and terbuthylazine, respectively) were too high. Nevertheless, the observed peak concentration for metolachlor at Lobith (20 ng/L) was close to the simulated value of 15 ng/L. For terbuthylazine, all simulated values at Lobith remained below the LOQ. This demonstrates at least that the concentrations were not overestimated. This contrasts with the results at Basel where the model predicted a maximum concentration 1.9 times the actual observation.

In our simulations, we have assumed that the compounds behave like conservative tracers without degradation or sorption taking place. Although this is not completely true, the travel times through the river network is so short that relevant dissipation can be expected to be negligible for the model compound considered in this paper (Honti et al., in preparation).

## 6 Discussion

### 6.1 Model performance

We presented here a series of predictions for herbicide and biocide concentrations in streams without any local calibration or model adaptations. In this sense, the results correspond to predictions in ungauged catchments covering tens of thousands of km$^2$ based on calibration catchments covering less than 30 km$^2$ in total. Despite the challenges that go with this task, the model validation demonstrated that the concentration levels could be predicted within a factor of two to four for 50 to 75% of the predictions. This is comparable to what has been observed for models predicting concentrations of micropollutants

from points sources (Johnson et al., 2008). The seasonality of the herbicide concentration peaks was well represented too. Overall, the results suggest that such a parsimonious model can be used as a meaningful screening tool to identify potential hotspots in river networks. Models of a similar degree of parsimony have been developed for point source pollution (e.g., Ort et al., 2009) but are largely lacking for compounds with rain-driven input dynamics.

Despite these positive aspects, one has to be clear about the limitations of the model and the resulting predictions.

Deficiencies are obvious when evaluating the performance metrics. The NSE or correlation coefficients obtained are low compared to values typically called satisfactory or good for hydrological models. However, our results need to be put into the context of comparable water quality studies dealing with diffuse pollution. As pointed out by Pullan et al. (2016) there is a lack of studies in this field reporting quantitative performance metrics such as NSE or r values. However, studies that do report such values demonstrate that the low NSE or correlation values of our work are in similar ranges of what others have

described. Table 5 and Figure summarise a selection of such findings from a number of model applications (e.g., SWAT, INC-P and others), which are much less parsimonious than the *iWaQa* model used in this study. This comparison indicates that model performance of water quality models achieved so far is generally considerably lower compared to what purely hydrological models can accomplished.

The fact that a parsimonious model such as the *iWaQa* model presented here was able to yield meaningful predictions

suggests that the model concept represents the effects of the major drivers controlling the degree and dynamic of biocide and



herbicide inputs into streams. It also indicates that these drivers remain constant over considerable spatial areas and that one can use findings from small study areas to extrapolate to larger basins as long as the first order controls do not strongly change. For the *iWaQa* model as implemented here this means that the herbicide input for example is mainly triggered by discharge events. However, in drier regions it may be possible that point sources play a dominant role (Müller et al., 2002).

In this case, the model concept had to be complemented to account for this input pathway (see Honti et al., 2017).

The observation that findings from small catchments can be extrapolated to larger areas in a meaningful manner may be considered a contradiction to earlier work where important spatial differences between herbicide loss rates within catchments were demonstrated (Doppler et al., 2014;Leu et al., 2010). However, the data suggest that spatial heterogeneity at small scales is averaged out at larger ones such that it does not dominate the large scale patterns.

## 6.2 Model limitations

Despite the positive aspects mentioned above, there are several (major) model limitations one has to be aware of. First, the parsimonious and empirical structure of the model requires compound-specific calibration. This generally implies that field data is available at the catchment scale with sufficiently well quantified input and output fluxes.

While this calibration step is also necessary for other (more complex) models there are also model limitations related to the

model structure. During calibration, we have noticed that the model was not able to fully represent the observed concentration peaks (see Table 4). This suggests that the model structure misses important processes that control concentrations during rainfall events. A possible candidate for such a process is drift deposition on roads and subsequent runoff during rainfall (e.g., Lefrancq et al., 2013). Interestingly, this systematic problem during the calibration phase was only partially observed during validation. Possibly this was due to the (much) larger scale of the validation catchments that

average over many temporally independent application events.

Other structural model limitations are too high herbicide background concentrations in some sub-basins, seasonal biocide concentration peaks that were not represented by the model or the lack of an isoproturon application in fall. These limitations were rather easy to identify but not very easy to solve. The herbicide application in fall for example is much more difficult to predict compared to the spring application because it not only depends on a single variable such as the temperature sum over

the year but it is also influenced by all the climatic variables determining the time of cropping of the previous crop and potential intercropping. For seasonal biocide patterns, we lack any information about biocide use on buildings that could explain the observed seasonality.

These examples demonstrate that the model limitations are often a mixture between too simplistic model structure and lack of input data. This agrees with the findings from the error models. The predictive uncertainty due to poorly identified

parameters only explain a small fraction of the deviations between observations and the deterministic model predictions in the calibration phase. The estimated uncertainty for the full error model however, covers most of the data. However, one should not overstate this observation because the fraction of uncertainty assigned to different sources depends on how the error model was formulated (Honti et al., 2014).

One could conclude that a more complex model was warranted to overcome such limitations. While this would be definitely

worth considering one should be aware of the severe limitations that come with the input uncertainty regarding the chemicals to be modelled. To illustrate this point, we have quantified the spatial and temporal density of input data needed for the model (Figure ). Compared to the drivers of the hydrological part such as precipitation the density of data on biocide and herbicide input into the system was orders of magnitude lower. While there is hourly precipitation data available on a 1 x 1





km$^2$ grid for the entire model domain we could only approximate the herbicide input based on average national sales data. For biocides, one had to rely on a single rough estimate per compound for the entire basin.

Given this level of input uncertainty, it comes as no surprise that the observed concentrations may be substantially over- or underestimated in a given subcatchment. The degree of mismatch between observations and simulations was still in a range that allowed to use the model as a screening tool for identifying potentially critical catchments in a basin. This was probably thanks to the widespread use of the selected model compounds. For less frequently used compounds, one can assume that the input estimates based on sales statistics would be even more uncertain due to e.g. region-specific application patterns. Accordingly, the predictive uncertainty would increase further.

## 7 Conclusions

Our findings suggest that even a very parsimonious model with a maximum of eight global parameters that need to be calibrated is sufficient to capture the key drivers and processes for diffuse agricultural herbicide and urban biocide losses reasonably well such as to predict level of peak concentrations within a factor of 2 to 4. This demonstrates that land use as a proxy for compound use, weather data for the timing of herbicide applications and discharge or precipitation as drivers for fast transport are first order controls for diffuse pollution for the compounds in our study area. The results further demonstrate that impact of these factors can be scaled spatially across at least four orders of magnitude (from < 3 km$^2$ to > 30'000 km$^2$).

At the same time the results also point to clear model deficiencies such as the simulation of background concentrations or the lack of the fall application of certain herbicides. Unfortunately, the analysis of model performance is limited by the lack of adequate validation data that have to combine reliable information on timing and amounts of the use of the chemicals and on concentrations in the streams. Progress in modelling and in measuring will remain closely coupled in this area and mutually benefit from each other.

Finally, it should be recognized that despite using a very parsimonious model, collecting the necessary input data and bringing it into a consistent form across a large water basin such as the Rhine is very time consuming. Hence, sharing model codes and even more importantly the required data will benefit the scientific community by not having to re-invent the wheel.

**Data and code availability**
The source code and the input data for the models has been placed to GitHub at https://github.com/moserand/crosswater.
The Supplement related to this article is available online at doi:10.5281/zenodo.556143.

**Author contributions**. CS designed the initial study design. AM, RS, FF, and CS continuously discussed and guided the project progress. DW, RS, and AM prepared the input data. DW and AM did most of the model coding with essential support from MH. AM and CS did most of the data analysis, figures where provided by RS, AM, and CS. AM and CS prepared the manuscript with contributions from all co-authors.

**Competing interests**. FF and CS are both on the editorial board of HESS. All other authors declare that they have no conflict of interest.





**Acknowledgements.**

The CrossWater project was financed by the Swiss National Science Foundation (Grant no. 406140-125866) and builds on previous work funded by the Swiss Federal Office for the Environment (contribution D. Wemyss). Hans-Peter Bader supported the project by regularly discussing important issues during the entire duration of the project. We like to thank

Mike Müller for getting us started in Python.

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





**Figures:**

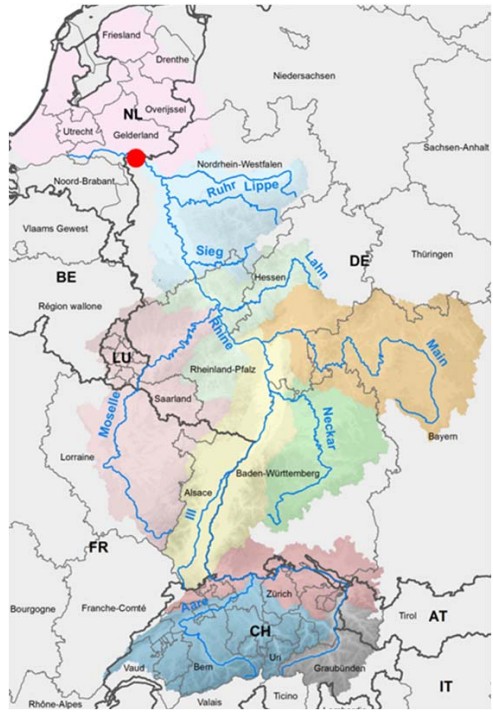

Figure 1: Map of the Rhine basin. The study area covers the part upstream of Emmerich indicated by the red circle. The different colours represent the sub-basins according to the International Commission for the Protection of the Rhine (ICPR) with the an additional distinction of the Aare basin in Switzerland. Base data: Vogt et al. (2007); Swisstopo (2007).





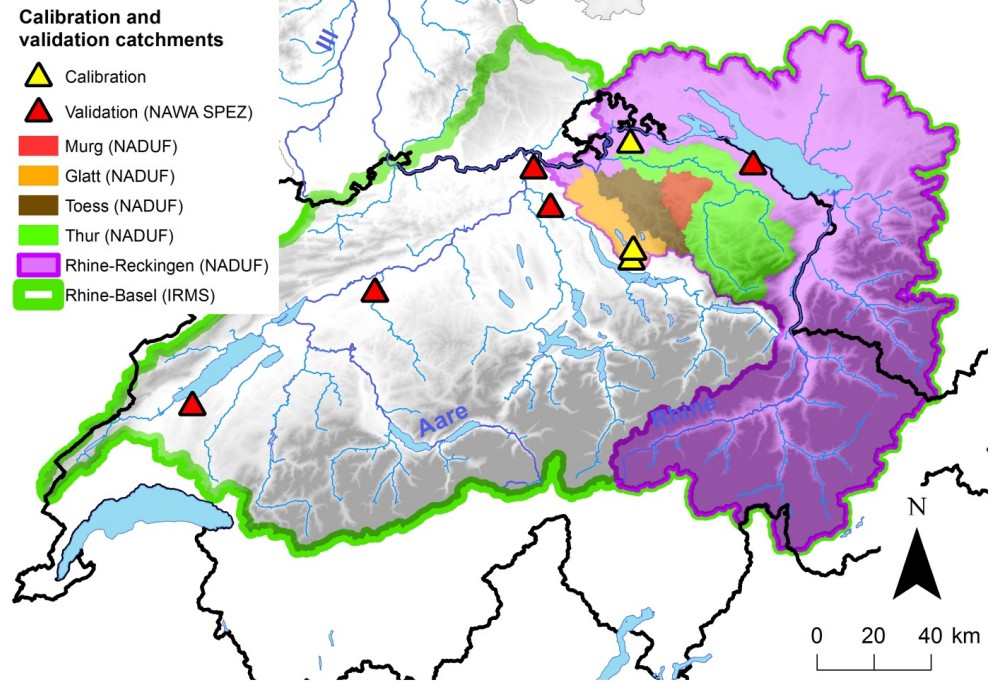

Figure 2: Calibration and validation catchments in Switzerland. Base data: Vogt et al. (2007); Swisstopo (2007).



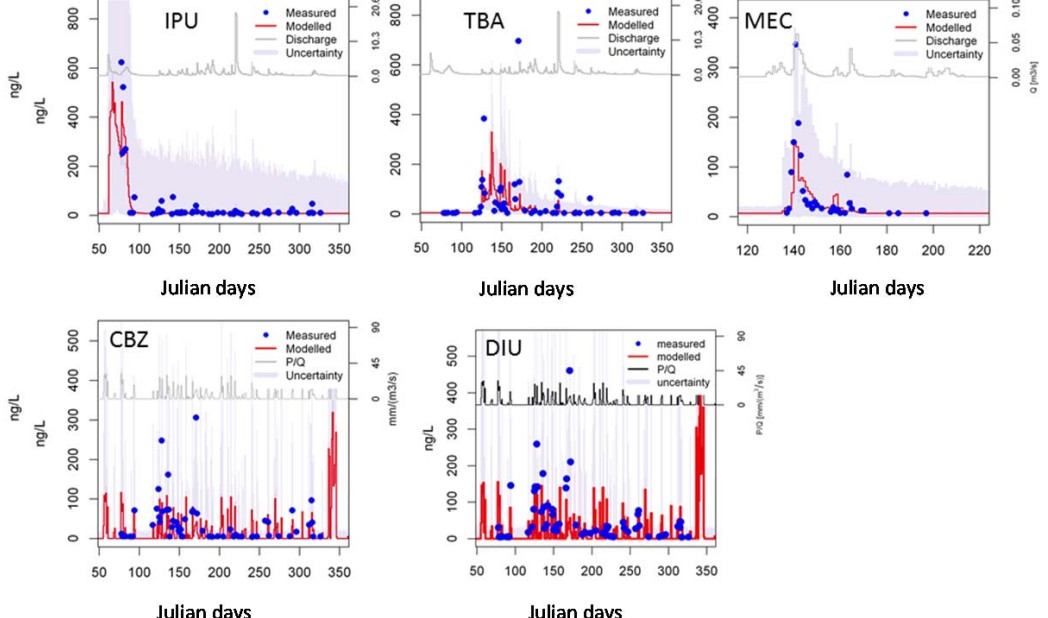

**Figure 3: Examples of the comparison between simulated and observed concentration time series during the calibration step for each compound. IPU: isoproturon (Mönchaltdorf), MEC: metolachlor (Summerau), TBA: terbuthylazine (Mönchaltdorf), CBZ:**

5  **carbendazim (Mönchaltdorf), DIU: diuron (Mönchaltdorf). The full set of calibrations is shown in the Supplementary Material (Fig. S7, S8).**





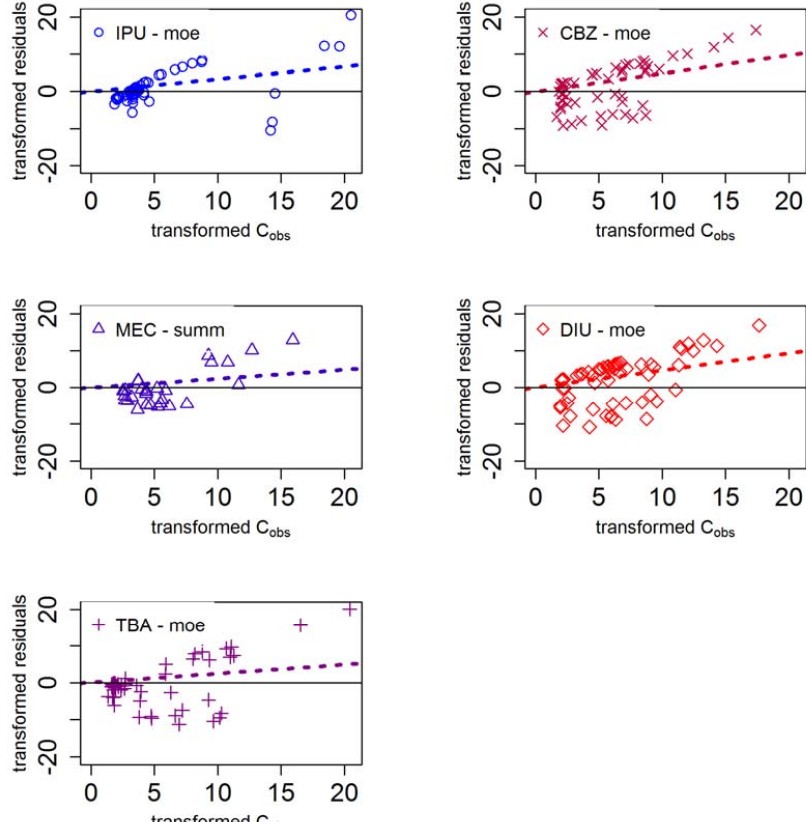

**Figure 4: Residuals (observed – simulated concentrations) for the model compounds during the calibration step. IPU: isoproturon,**

**MEC: metolachlor, TBA: terbuthylazine, CBZ: carbendazim, DIU: diuron. Moe: Mönchaltdorf, summ: Summerau.**




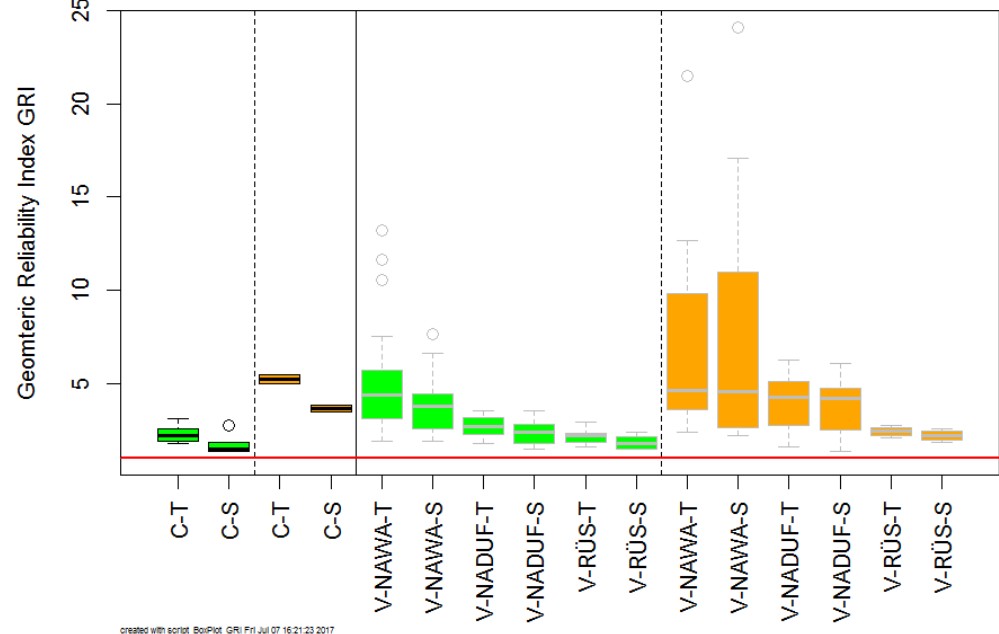

**Figure 5: Overview about the overall predictive power to simulate the concentrations levels during the calibration and validation phase as quantified by the geometric reliability index (GRI). A value of 1 (red horizontal line) indicates a perfect match; the larger the value the stronger the deviation. C: calibration; V: validation; -T: evaluation of concentration time series; -S: evaluation of cumulative concentration distributions (sorted according to size); green: agricultural herbicides; orange: dual use (urban and agricultural) biocides.**





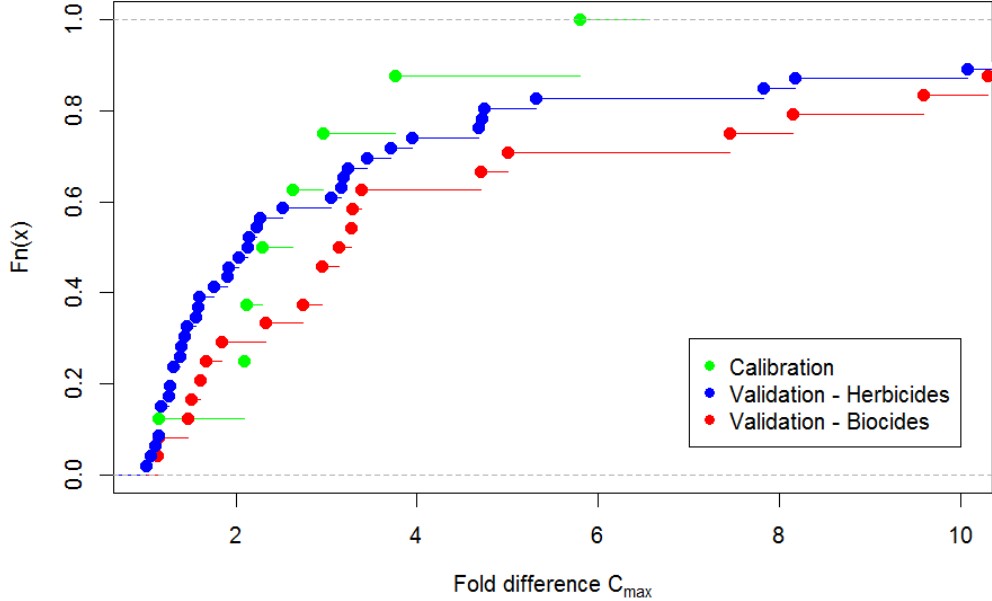

**Figure 6: Cumulative distribution of the fold difference between observed and simulated concentrations $C_{max}$ of all compounds during the calibration and validation phase.**

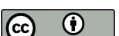


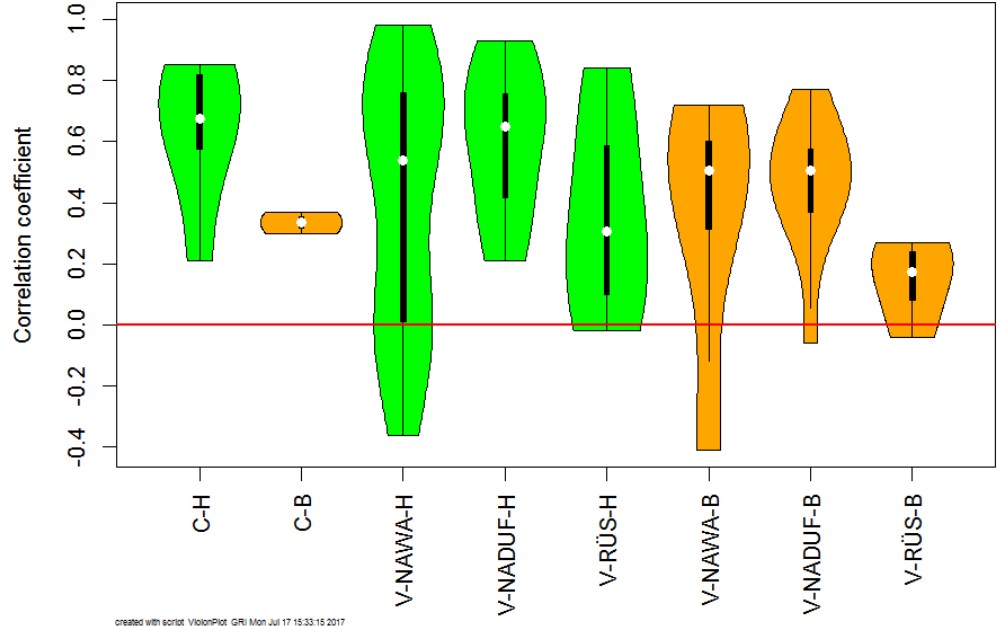

**Figure 7: Violin plots of the Pearson correlation coefficients between simulated and observed concentrations for (H) herbicides**

5    **(green) and (B) biocides (orange) during (C) calibration and for the different (V) validation data sets.**





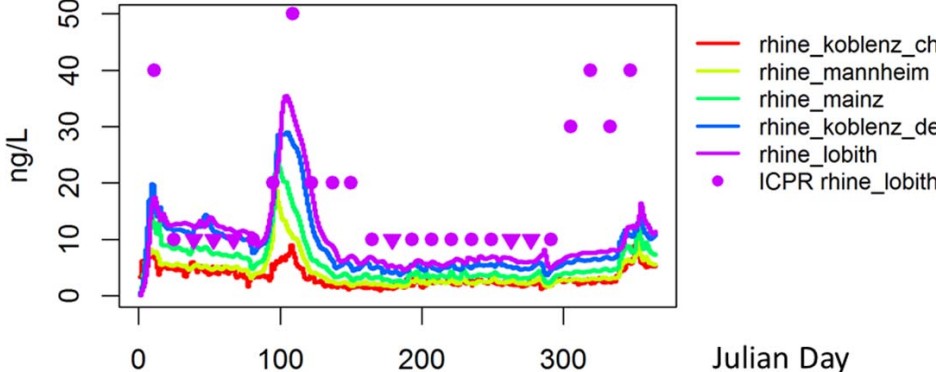

**Figure 8: Comparison of predicted isoproturon concentrations along the River Rhine for 2011 compared to the observations at the measuring site at Lobith.**





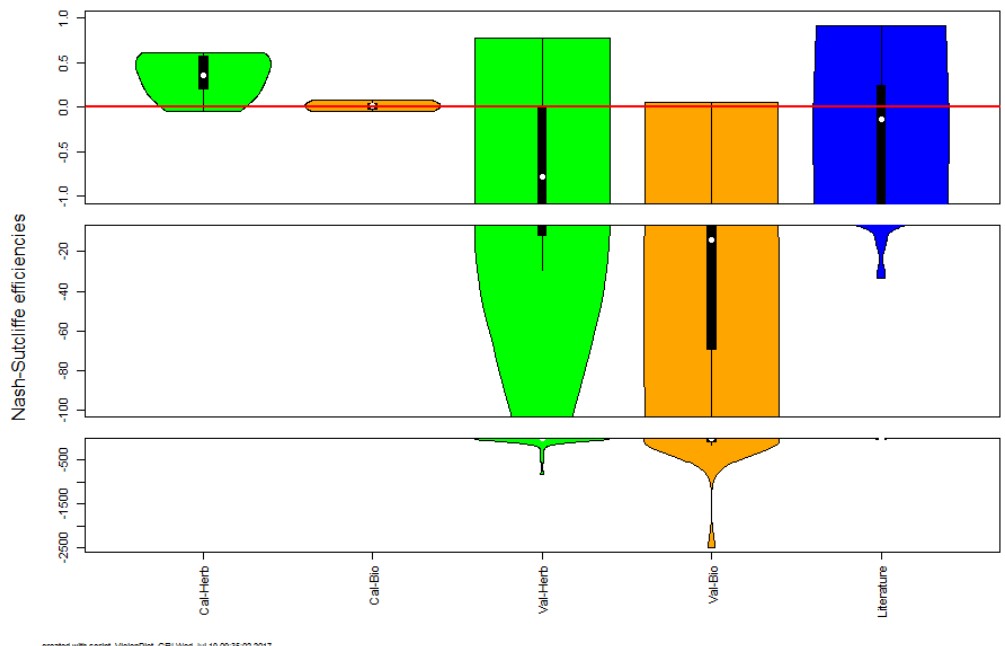

created with script_ViolonPlot_GRI Wed Jul 19 09:35:02 2017

**Figure 9: Comparison of Nash-Sutcliffe efficiencies in this study for herbicides (green) and biocides (orange) during calibration (C) and validation (V) with values from the literature for diffuse pollutants (herbicides, different P-forms; sources: see Table 5).**




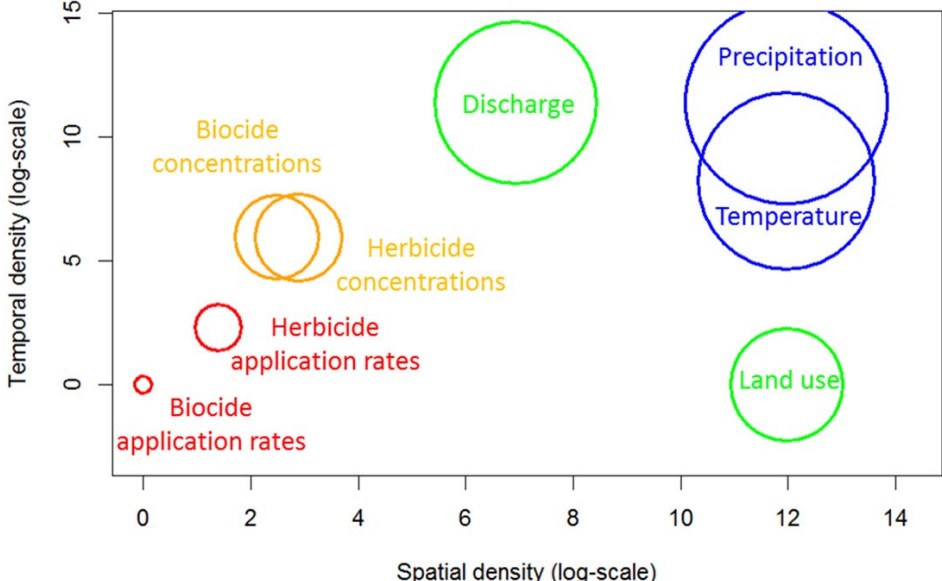

**Figure 10: Spatial and temporal density of different input variables. The different colours represent different categories of input**

**quality.**





**Tables**

Table 1: Global model parameters.

| Abbreviation | Name | Description | Specificity | Model part |
|---|---|---|---|---|
| $\rho$ | Initially available herbicide fraction | Fraction of the applied herbicide mass initially available for transport | compound | deterministic |
| $k_{w-s}$ | Sorption rate | Fraction of the dissolved herbicide mass getting sorbed to the soil matrix per unit of time | compound | Deterministic model |
| $k_{s-w}$ | Desorption rate | Fraction of the sorbed herbicide mass getting desorbed per unit of time | compound | Deterministic model |
| $k_{deg}$ | Degradation rate | Rate constant of the first order degradation | compound | Deterministic model |
| $C_{back}$ | Background concentration | Constant background concentration, proportional to the areal fraction of the relevant crop in the subcatchments | compound | Deterministic model |
| $\epsilon$ | Herbicide loss rate | Loss rate per unit discharge and available herbicide mass | compound | Deterministic model |
| $T_{obj}$ | Temperature objective | Cumulative temperature sum required to start herbicide application on a crop | crop | Deterministic model |
| $\beta$ | Biocide loss rate | Loss rate per unit precipitation and available biocide mass | compound | Deterministic model |
| $\mu$ | Scaling factor | Factor for scaling the model error term proportional to the subcatchment-specific herbicide input | compound | Error model |
| $\sigma_{error}$ | Standard deviation of the error model | Relative standard deviation of the total model error | compound | Error model |





**Table 2: Characterization of calibration and validation catchments**

| Catchment/ River | Abbr. | Reference | Year | Area [km$^2$] | agricultural land use [km$^2$] | Housing footprint [km$^2$] | population |
|---|---|---|---|---|---|---|---|
| **Calibration** | | | | | | | |
| Ossingen | oss | Doppler et al. 2012 | 2009 | 1.2 | 1.1 | - | - |
| Summerau | sum | Freitas et al. 2008 | 2003 | 0.5 | 0.04 | - | - |
| Mönchaltdorf | moe | Wittmer et al. 2010 | 2007 | 24.6 | 4.7 | 0.5 | 12'000 |
| **Validation** | | | | | | | |
| Furtbach | fch | NAWA SPEZ | 2012 | 31 | 14 | 1.6 | 31'570 |
| Limpach | lch | NAWA SPEZ | 2012 | 74 | 43 | 1 | 7'560 |
| Mentue | mnt | NAWA SPEZ | 2012 | 100 | 42 | 1 | 9'300 |
| Salmsacher Aach | smr | NAWA SPEZ | 2012 | 54 | 33 | 1.7 | 17'326 |
| Surb | srb | NAWA SPEZ | 2012 | 68 | 36 | 1.4 | 22'780 |
| Thur | thr | NADUF | 2009 | 1'735 | 873 | 33 | 403'028 |
| Toess | tss | NADUF | 2009 | 432 | 175 | 11 | 197'032 |
| Glatt | glt | NADUF | 2009 | 413 | 183 | 20 | 405'702 |
| Murg | mrg | NADUF | 2009 | 212 | 118 | 5.3 | 68'145 |
| Rhine-Reckingen | rhn | NADUF | 2009 | 14'721 | 5'261 | 175 | 2'946'907 |
| Rhine-Basel | irms | IRMS | 2010/11 | 35'899 | 12'009 | 503 | 7'786'398 |



**Table 3: Metrics used for quantifying model performance.**

| Metric | Abbreviation | Description |
|---|---|---|
| Nash-Sutcliffe Efficiency | NSE | $NSE = 1 - \dfrac{\sum_{i=1}^{n}(o_i - m_i)^2}{\sum_{i=1}^{n}(o_i - \bar{o})^2}$ |
| Pearson correlation coefficient | r | $r = \dfrac{\sum_{i=1}^{n}(o_i - \bar{o})(m_i - \bar{m})}{\sqrt{\sum_{i=1}^{n}(o_i - \bar{o})^2}\sqrt{\sum_{i=1}^{n}(m_i - \bar{m})^2}}$ |
| Percent bias | PBIAS | $PBIAS = 100 \times \dfrac{\sum_{i=1}^{n}(m_i - o_i)}{\sum_{i=1}^{n} o_i}$ |
| Relative root mean square error | RRMSE | $RRMSE = \dfrac{\sum_{i=1}^{n}|m_i - o_i|}{n\, \sigma_{obs}}$ |
| Geometric Reliability Index (cumulative distribution) | GRI (GRI_sorted) | $GRI = \dfrac{1 + \sqrt{\frac{1}{n}\sum_{i=1}^{n}\left(\frac{m_i - o_i}{m_i + o_i}\right)^2}}{1 - \sqrt{\frac{1}{n}\sum_{i=1}^{n}\left(\frac{m_i - o_i}{m_i + o_i}\right)^2}}$ |
| Relative difference between maximum concentration | $\Delta_{Cmax}$ | $\Delta_{c_{max}} = \dfrac{C_{max}^{sim} - C_{max}^{obs}}{C_{max}^{obs}}$ |
| Fold difference between maximum concentration | F.diff | $F.diff = \begin{cases} \dfrac{C_{max}^{sim} - C_{max}^{obs}}{C_{max}^{obs}} & C_{max}^{sim} > C_{max}^{obs} \\ \dfrac{C_{max}^{obs}}{C_{max}^{obs} - C_{max}^{sim}} & C_{max}^{sim} < C_{max}^{obs} \end{cases}$ |



**Table 4: Over- or underestimation of maximum concentrations (site-compound combinations) in percentage of the observations.**

**For the herbicides only the peaks during spring application were considered. IPU: isoproturon, MEC: metolachlor, TBA:**

**terbuthylazine, CBZ: carbendazim, DIU: diuron.**

|  |  | IPU | MEC | TBA | CBZ | DIU |
|---|---|---|---|---|---|---|
| Calibration | Mönchaltdorf | -13 | -51 | -53 | -62 | -66 |
|  | Ossingen | -71 | - | -83 | - | - |
|  | Summerau | - | -58 | - | - | - |
| Validation SPEZ | Furtbach | 6 | -10 | 431 | 61 | 715 |
|  | Salmsacher |  |  |  |  |  |
|  | Aach | 114 | 17 | 1898 | 229 | 1201 |
|  | Surb | 123 | -53 | 56 | -32 | 859 |
|  | Limpach | 103 | -14 | 17 | -57 | 2772 |
|  | Mentue | 2405 | 45 | 43 | 84 | 370 |
| Validation NADUF | Thur | -9 | -47 | - | -57 | 91 |
|  | Rhine |  |  |  |  |  |
|  | Reckingen | 22 | 20 | - | -65 | 70 |
|  | Murg | -42 | -61 | - | -97 | 221 |
|  | Toess | -35 | -37 | - | 458 | 265 |
|  | Glatt | 92 | -45 | - | 4 | 789 |
| Validation IRMS | Rhine Basel | -67 | -60 | 368 | 239 | 931 |





**Table 5: Examples of reported Nash-Sutcliffe efficiency values and Pearson correlation coefficients between observations and simulations reported for a selection of water quality modeling studies.**

| Reference | Compound | C/V | NSE | r |
|---|---|---|---|---|
| (Bannwarth et al., 2014) | Atrazin | C | 0.92 | - |
| | | V | 0.61 | - |
| | Chlorothalonil | C | 0.67 | - |
| | | V | 0.28 | - |
| | Endosulfan | C | 0.86 | - |
| | | V | 0.31 | - |
| (Parker et al., 2007) | Atrazine | C | -0.18/-1.03/-3.50[†] | 0.12/0.30/0.64[†] |
| | Metolachlor | C | -0.84/-3.53/-33.4[†] | 0.14/0.46/0.57[†] |
| | Trifluralin | C | -30.2/-16.9/-3.2[†] | -0.16/0.35/0.14[†] |
| (Boulange et al., 2014) | Mefenacet | S | 0.65/-9.72/-14.7[‡] | 0.78/0.87/0.92[‡] |
| (Holvoet et al., 2008) | Atrazine | C | 0.66 | - |
| Holvoet 2007 | Chloridazon | C | -0.67[#] | 0.44[#] |
| (Jackson-Blake et al., 2015) | Suspended sediment | C | 0.16/0.39/0.21/0.02[*] | 0.63/0.83/0.64/0.21[*] |
| | TDP | C | 0.24/0.04/-0.20/-0.60[*] | 0.83/0.68/-0.05/0.27[*] |
| | Different P forms | C | 0.06/-0.14/-0.60/-0.42/-1.15/-4.18/0.19/-0.08/-0.74/0.08[$] | |

[†]: values for three different models

