# Peer review of "Modelling biocide and herbicide concentrations in catchments of the Rhine basin"

_Hydrology and Earth System Sciences, 2017_

## Referee Comment (RC1) · Anonymous Referee #1 · 1 Feb 2018

General comments

Developping and testing the limits of a parsimonious model of micropollutants transport from catchment to river at various scales is relevant both for stakeholders and for the scientific community to address the degree of simplification required/able to capture the micropollutants patterns. The key concept of this approach (link the load of micropollutants to the discharge and/or the rain) is not new but the approach to validate at small scale with European databases the load then upscaling the approach at larger scale reaching the main part of the Rhine catchment is new. The spatial preprocessing of existing European data to improve the information for the subcatchment can also be underlined. The state of the art on the different components covered by the work is well presented with relevant references. The model development, main hypotheses and calibration/validation/transposition steps are clearly presented. However, a scheme summarizing spatial and temporal discretization with associated processes across scale (calibration then validation catchments and full Rhine scale) is clearly missing in the main text. A very simple scheme (focused on subcatchment delineation) is presented in SI but can't play this summary goal.

The results are clearer for herbicides than for biocides. The reader discovered different hypotheses that reduce progressively the extent of the biocides loads ant transport modelling at larger scale. I wonder if a focus on herbicides only, should not be better and stronger. I detailed below specific comments and corrections

P1 L30: check homogeneity (S-metolachlor and metolachlor are used in the text)

P2 L28: Missing dot at the end of a sentence "2013) One"

P3 L13-14: original aspect of this work +add that it's a daily or hourly time step

P3 L23: all the basin? just after it's mentioned that only the basin upstream of the station Emmerich am Rhein is covered by this study, precise

P4 L6: general comment: An overall scheme of the model should be relevant to improve understanding of the spatial links between objects, considered processes and links between AWaQa and AQUASIM (what's happened in the sub-catchment scale and then in the river, degradation, . . . ) The appendix A2 is one aspect of the discussion but it not covers the processes.

P5 L8 and L9: in the equations 2 and 3 Kdeg has to be used instead of Kd to be homogeneous with the rest of the paper

P5 L11: not clear at this stage how the available fraction is link to rainfall

P5 L21: interesting but why 1/14? Expert panel, reasonable fractionation? Sensitivity of the model to this fraction?

Does it means that 14 days are required to consider 100% of application? Can this

hypothesis impact peak modelling due to dilution of input signal?

P6 L23: no rainfall dependency?

P6 L34-35: not clear for me, does it mean that any transformation is considered in the river routing model?

P7 L8: should be interesting to mention in the abstract and in the introduction...

P7 L11-12: is it possible to mention an unpublished paper in HESS?

P7 L17: I think it is the table S2 (in the appendix A4) and not the table S4.

P7 L20: not easy to understand even if it's describe below... a scheme should be relevant to improve understanding.

P7 L25: what % considered considering strahler order less than five?

P7 27: Explain briefly the area ratio method to help the reader at this stage

P7 31: Same comment, explain briefly the map-correlation method to help the reader at this stage

P8 L16-18: not clear for me! -6.5° per km (in z?)

P8 L21: Expected evolution if 2000 is the reference? Impact of rotation of crops (spatial difference between years)?

P9 L17: Diuron was also used in vineyard still 2008 in France. Is it the same in Germany and Switzerland? Expected impact of missing the agricultural uses in this analysis?

P9 L24-29: not clear for me: range of available data? 2008-2012 so 5 years for all the study site excepted for Lorraine? Right? You have to mention the Appendix A5 (figure S3 and S4) to visualize the spatial variability. The reader discovers in the caption of the figure S4 (appendix A5) that the Diuron pattern was just a copy of the carbendazim map. Is it correct? I do not find any discussion on that in the text.

P9 L34: does any other sources can (even partially) validate this hypothesis?

P10 L22: "application season" Perhaps explain why. . . to take into account the fact that during application period difference can occur between real applications (unknown) and modelled application (splitted with 1/14 depending of weather windows).

P10 L22: "error-scaling function" try to better introduce this function, why and how, it's not very clear for the reader.

P10 L30: the text and the equation 13 are not supported by any reference

P11 L10: studies are mentioned to assess the prior distribution of ïĄě, which one? Wittmer et al (2010) mentioned in L26?

P11 L28: I suggest to mention the table S4 and S5 (Appendix A8) after "of the priors for ïĄě and ïĄć"

P12 L28: "larger rivers" (Rhine, Aare;" I suggest to mention all main tributaries or use "such as"

P12 L31: The fact that biocide was finally not modelled at the Rhine scale (due to lack of biocides export coefficient in France and Germany) should be precise clearly in the abstract and introduction by differencing the two scales (calibration/validation in the Switzerland scale of herbicides and biocides, and only extrapolation of herbicides at the Rhine scale).

P13 L20: I suggest to link here the table S6 and S7 no called in the main text.

P13 L27-28: this sentence needs to be followed by some hypotheses for this bi-modal pattern, especially if physical explanations can help to improve the model for biocides release.

P13 L 32: I can't see how two clusters can be derived from the figure 4. Could you clarify this point?

P13 L39: because the GRI is probably less known compared to the NSE criteria, it could be relevant to provide quality thresholds to consider poor, acceptable, good and very good capabilities of the model.

P14 L3: could you provide in SI Table S9 the unit of RRMSE (%?) Its not clear in SI if it's really in % (very low value if in %). The table S9 is not used to support quality of the model in the main text.

P14 L2: what could be the interest to reproduce the cumulative concentration (observed and simulated sorted) if the dynamic (timing) is not correctly capture?

P14 L16: how the reader can derive this information? Which tables or figures supports this statement?

P14 L24-25: If I well understand, you observed more dissipation than prior estimation. Does it mean that missing dissipation pathways (leaching to deeper groundwater) could be counterbalance in the model by more sorption (model structure error)?

P15 L2-3: do you mean cumulative distribution or chemogram (dynamic evolution of concentration)?

P15 L4: Could you provide hypothesis or compare this behaviour whith other similar approach (calibration in small scale and validation at large range of scale)?

P15 L15: do you have hypothesis for this over-estimation? Could this overestimation in the validation sites be due to an overestimation of the diuron release in the calibration sites owing to agricultural use neglecting?

P15 L26: Could you provide figures or tables (in the main text or SI) to help the reader to understand the differences between the 2 routing methods?

Perhaps it could be relevant to sort the results depending of the catchment area to underline the threshold from which the full routing model improve the concentration prediction.

P15 L39-40: you mentioned previously that you did not simulate the biocide due to a lack of the input database for Germany and France.

P15 L2: missing figure number! Figure 8, I guess

P15 L4: why applications during fall were not considered? Isoproturon is usually applied in October on winter wheat

P16 L16: Probably not at the larger scale! It's probably a strong hypothesis for which scenarios with and without river processes (degradation and sorption) could be tested with available dataset (DT50 water...)

P16 L15: same comment as previously for unpublished paper (possible in HEES?)

P22 L6: Figure 1 : I suggest a modification of the titre: the study area covered the Rhine viver upstream the Emmerich discharge gauge (red circle)

I suggest also to delineate more clearly the Rhine basin with bold line

P24 L3: Figure 3: I suggest to express concentration in $\mu$g/L to be able to better link them to EU drinkable threshold (0.1 $\mu$g/L)

I also suggest to add application dates to see interplay between application and rainfall calendars

For biocides, P/Q is not easily understandable, is it a ratio? The sentence (Line 26-27, page 13) that biocide concentration follows rainfall patterns can not easily be derived from this figure), especially at the end because legend hides the rainfall/discharge dynamic (probably high)

Finally, on the left you have to remove one ng/L (two times in the figure)

P16 L25: I agree for herbicides, but it could be remembered that seasonal peaks of biocides were not well represented. This weakness seems not to be discussed in the paper. Which types of processes can explain a seasonal variation of biocides exports

from roofs and walls?

P16 L27: the authors argue that iWaQa can help to identify potential hotspots in river network. I'm not convince according the way that herbicides loads is calculated. Could the authors more clearly underline the strengths and weaknesses of this model to identify hotspot according conceptual structure and processes taken into account? If I well understand, the inputs are derived from administrative data and can not explain extreme applications and associated peak of concentration at small scale.

P16 L35: missing number of figure Table 5 and Figure . . .?

P17 L5: modify the way that the reference is called. I suggest "as discussed in Honti et al. (2017)

P17 L21 to27: I suggest to order the different elements (1) too high herbicide. . ., (2) seasonal biocide . . . and (3) the lack of an isoproturon application. . . In the following sentences, I only see discussion on the points 3 and 2 and not for the point 1. Reorganize this section.

P17 L29: "This agrees with the findings from the error models", I suggest to link here the tables in SI providing range of error.

P17 L37: missing number for the figure mentioned

P25 Figure 4: even if the calculation is remembered in the caption, it could be relevant to add in the different box of this figure, arrows and captions underlining that if residual > 0 the model underestimated concentration and the opposite if < 0.

I also suggest to move this figure in the SI.

P26 L2: I suggest to explain that the catchments are sorted by size (with an arrow and associated surface to better support the sentence (Line 4-5, page 15)

P28 Figure 7: move to SI

P30 Figure 9: move to SI

---

## Referee Comment (RC2) · Anonymous Referee #2 · 5 Mar 2018

**Manuscript Number:**     HESS-2017-628

**Authors:**     Moser, A. et al.

**Manuscript Title:**     Modelling biocide and herbicide concentrations in catchments of the Rhine basin

General Comments:

The authors have used the parsimonious model *iWaQa* for predicting temporally resolved concentrations of pesticides (from agricultural runoff) and biocides (from diffuse loss from urban areas) in large parts of the Rhine river watershed. The overall goal of the paper was to calibrate the described model and evaluate its prediction capability in a case study in the Rhine basin. I especially appreciate that the authors clearly mention the limitations of the model approach and the time-consuming data collection process often hampered by insufficient data availability.

The manuscript is well-structured and written in good English. Description of the model framework as well as its parameterization and calibration is straightforward and despite its complexity well understandable to me. The line of reasoning in the discussion is mostly convincing to me, but presentation of results is supported by a number of figures and tables that are not always easily understandable and not all are meaningful to me in its current form (see specific comments).

**Nevertheless, I can recommend the manuscript to be accepted for publication after minor revision according to the comments below.**

Specific comments

- Literature with respect to other related model work in this field is not sufficiently reviewed. References are sometimes unfortunate, e.g. MONERIS is cited not with original papers but with a side application and reference to GREAT-ER is outdated (see Kehrein et al., 2015). The EU approach (FOCUS model suite) is neglected as is the DRIPS model (Röpke et al., 2004).

- The assumption of general presence of persistent pesticides in baseflow and groundwater (page 6) is somehow terrifying to me, since prevention of elevated concentration levels in groundwater is the main driver for research in this field. Background concentrations determined in the calibration step need to be explicitly compared to the large number of data available. Fortunately, many of these data points are small and below critical limits. These data allow for defining an upper bound for the calibrated background concentrations.

- Many German cities situated in the Rhine catchment use combined sewer systems, whereby a significant fraction of urban surface runoff is directed not instantaneously into rivers (page 6), but to the nearest sewage treatment plant. While the substance-specific load reduction during wastewater treatment is included in the loss rate $\beta$, it should be discussed whether retardation of transport into the receiving waters for the hydraulic retention time in the sewer system and the wastewater treatment plant could have an effect on the timing of the biocide concentration peaks in the simulations.

- I think that some figures in the main text are not important enough to justify their presence. Moreover, the meaning of some figures is hardly recognizable from the discussion. I strongly encourage the authors to shift some of the figures to the supporting information. In the main text, I would focus the discussion on the most important points supported by meaningful and easily understandable figures with clearly labelled axes.

  - Figure 4: I could not find a description of the transformation step used to generate the data displayed here. Both axes are displayed in dimensionless scale, but I have no idea what that means. It remains also unclear, what the dashed line represents and why the Figure points to systematic deviations between observed and modelled concentrations (page 13, line 31)

- Figure 5: It is not possible to extract the numbers for the GRI values given in the text (page 13/14) from the graph. For me, the range shown in the graph is larger with values up to 5 for the herbicides. Meaning of vertical lines is unclear and the error bars (grey) unreadable. Colour of second section (C-T, C-S) is not identifiable and it is confusing that all categories appear twice on the category axis.
  - Figure 10: Scaling of the axes is unclear (log-scale of what?)
- Model performance for biocides is poorer than for pesticides (page 14, line 2), which should be discussed in the light of the different number of calibrated parameters for herbicides (seven/nine with error function) and biocides (just one). Either the biocide model needs more detailed process descriptions or the herbicide model is over-parameterized mutually levelling off uncertainties introduced by the parameters.

Minor comments
- pathways, wastewater (page 2)
- … where M(t) [g d$^{-1}$] **is** the mass ratio … (page 5)
- … temperature sum models is used (page 5)
- … of the pronounced concentrations peaks (page 15)
- Number of Figure missing (page 16, line 2)
- … for the model compounds considered in this paper (page 16)
- …micropollutants from points sources (page 16)
- …purely hydrological models can accomplish (page 16)
- Number of Figure missing (page 17, last but one line)

---

## Referee Comment (RC3) · Anonymous Referee #3 · 6 Mar 2018

The authors present an extensive modeling study on pesticide transport in the Rhine basin. The paper including the supplement is very long (as is the river Rhine). It builds on quite an impressive amount of work and is well written, although not everything is explained in full detail. The latter is probably unavoidable with such kind of studies. I appreciate that the authors provide there input data and model code.

The basic idea is to set up a simple, parsimonious model, to calibrate it with detailed data (from Switzerland, where a number of very good, detailed studies were carried out in the past) and predict transport on large scale. The approach is actually quite successful. Modeled concentrations partly deviate from measured concentrations by up to a factor of 5 or so, but we should also not forget that the concentrations are low. It is clear that the model can be improved when better input data become available (if

ever).

I have only one major point to make. I refer to the last paragraph of section 5.1. The authors used a (lumped) modeling approach in which sorbed phase and solution phase concentration are expressed in total masses in the system (watershed). The model was parameterized with laboratory data from a study by Freitas et al. 2008 (reference missing), but it is not explained how. Was the different soil to solution ratio in the lab considered? If the model is directly applied to the lab data ($M_s$=sorbed, $M_w$=dissolved), it would yield too low partitioning coefficients, what is indeed what the authors observed (section 5.1, last paragraph). This is so because, as a physicochemical fact, the sorption isotherm is independent of the soil to solution ratio as has been shown many times in the literature. In the lab, the mass in solution is much higher (by a factor of 5 or so, depending on the soil to solution ratio used). Taking it differently, sorption isotherms measured in the lab cannot be applied to a system (watershed) without making an assumption about the soil to solution ratio in it.

Unfortunately, my print-out version did not contain page numbers.

page 1, line 17

to what degree

At some locations in the text, references to figures lack the number (figure [empty]).

eq 1

$M(t)$ is undefined (cf. eq 8)

eq 8

in eq 2 and 3, $M_a$ was defined as the mass applied to the catchment the mass present in the catchment should be $M$ as in the LHS of eq 1

Figure 4

how were the residuals transformed?

Figure 5

Overview of

Figure 6

I did not get the rational behind the fold difference. This should be explained in the text and in the figure caption.

Figure 7

What is the violin about? Is the envelope a standard error? Explain in the text and in the figure caption.

Figure 9

Explain the envelope?

Figure 10

I have no clue what this figure is about. Explain the axes, assign units. Explain properly or delete the whole thing.

Table 1

Did you properly explain the scaling factor $\mu$? I think no.

Table 2

Explain the abbreviations (like NADUF).

Table 3

Explain the symbols in the caption.

Table 4

Explain abbreviations.

Table 5

C and V should be defined. Atrazine

***Summing up, the tables and figures should be self-explaining.

---

## Author Comment (AC2) · 29 Mar 2018

Response to Reviews

To all reviewers

Thanks for the detailed and constructive comments. Below we added our responses (in red) to each single point.

On behalf of all co-authors.

Christian Stamm

Anonymous Referee #1

General comments

*Developing and testing the limits of a parsimonious model of micropollutants transport from catchment to river at various scales is relevant both for stakeholders and for the scientific community to address the degree of simplification required/able to capture the micropollutants patterns. The key concept of this approach (link the load of micropollutants to the discharge and/or the rain) is not new but the approach to validate at small scale with European databases the load then upscaling the approach at larger scale reaching the main part of the Rhine catchment is new. The spatial preprocessing of existing European data to improve the information for the subcatchment can also be underlined. The state of the art on the different components covered by the work is well presented with relevant references. The model development, main hypotheses and calibration/validation/transposition steps are clearly presented.*

Thanks for the positive feedback.

*However, a scheme summarizing spatial and temporal discretization with associated processes across scale (calibration then validation catchments and full Rhine scale) is clearly missing in the main text. A very simple scheme (focused on subcatchment delineation) is presented in SI but can't play this summary goal.*

We agree that such a scheme would be useful and will include an improved version of the current figure in the SI into the main text.

*The results are clearer for herbicides than for biocides. The reader discovered different hypotheses that reduce progressively the extent of the biocides loads ant transport modelling at larger scale. I wonder if a focus on herbicides only, should not be better and stronger.*

We agree that the data basis for modelling biocides is substantially poorer than for herbicides. However, we consider biocides as relevant compounds and find it thus important to report the modelling challenges and thereby illustrate the limitations when simulating such compounds.

*I detailed below specific comments and corrections*

*P1 L30: check homogeneity (S-metolachlor and metolachlor are used in the text)*

Thanks for the careful reading. We will check for consistency, use consistent terminology and make it explicit if one has to switch to the other expression because data do not allow the specific use of S-metolachlor.

*P2 L28: Missing dot at the end of a sentence "2013) One"*

Will be corrected.

*P3 L13-14: original aspect of this work +add that it's a daily or hourly time step*

We will add the temporal resolution.

*P3 L23: all the basin? just after it's mentioned that only the basin upstream of the station Emmerich am Rhein is covered by this study, precise*

We will mention here that the basin upstream of Emmerich is covered.

*P4 L6: general comment: An overall scheme of the model should be relevant to improve understanding of the spatial links between objects, considered processes and links between AWaQa and AQUASIM (what's happened in the sub-catchment scale and then in the river, degradation, … ) The appendix A2 is one aspect of the discussion but it not covers the processes.*

As mentioned above, we agree and will add a corresponding scheme.

*P5 L8 and L9: in the equations 2 and 3 Kdeg has to be used instead of Kd to be homogeneous with the rest of the paper*

Thanks for noticing. This will be corrected.

*P5 L11: not clear at this stage how the available fraction is link to rainfall*

This fraction is not linked to rainfall but describes the effect of immediate sorption in contrast to the slower exchange processes described by Eq. [3]. We try to rephrase such to avoid misunderstandings.

Suggested sentence: "… available for transport, such that it can be directly mobilised when it rains."

*P5 L21: interesting but why 1/14? Expert panel, reasonable fractionation? Sensitivity of the model to this fraction? Does it means that 14 days are required to consider 100% of application? Can this hypothesis impact peak modelling due to dilution of input signal?*

Yes, the total application mass is distributed over 14 days. This reflects the fact that pesticides are not applied on all fields in a subcatchment on the same day. With applications taking place only during dry days, the total application period lasts 3 – 4 weeks which is reasonable based on our field experience.

These assumptions have of course an influence on simulated peak concentrations because they only reflect an approximation to the actual temporal application pattern, which is unfortunately not known.

P6 L23: no rainfall dependency?

Yes, the load is proportional to rainfall - the last term (P(t)) represents daily precipitation.

P6 L34-35: not clear for me, does it mean that any transformation is considered in the river routing model?

Yes, the load aggregation models neglects transformation processes while the routing procedure with Aquasim in principle could.

P7 L8: should be interesting to mention in the abstract and in the introduction...

We will mention the coupling with Aquasim in both parts.

P7 L11-12: is it possible to mention an unpublished paper in HESS?

We will skip this reference and the one on page 16

P7 L17: I think it is the table S2 (in the appendix A4) and not the table S4.

Thanks for noticing. This will be corrected.

P7 L20: not easy to understand even if it's describe below... a scheme should be relevant to improve understanding.

We will add a schematic representation for better illustration.

P7 L25: what % considered considering strahler order less than five?

We will add the respective information.

P7 27: Explain briefly the area ratio method to help the reader at this stage

The area ratio method simply assumes that discharge scales proportional to catchment area (see e.g. Hirsch 1979). We will add this explanation to the sentence: "…, which assumes that discharge scales proportional to catchment area." .

P7 31: Same comment, explain briefly the map-correlation method to help the reader at this stage.

We will add a sentence for more explicit explanation: "Selection of the reference stations is based on the map-correlation method from Archfield and Vogel. (2010). This geostatistical method calculates the correlation between discharge time series at observed stream gauges and estimates the station with the most correlated discharge at the ungauged catchment based."

P8 L16-18: not clear for me! –6.5 per km (in z?)

We will rephrase ("From these altitude deviations, temperature values were corrected based on a temperature decrease by -0.0065°C/m altitude increase".)

P8 L21: Expected evolution if 2000 is the reference? Impact of rotation of crops (spatial difference between years)?

This is a very important point and there are two aspects to this question. The first relates to the long-term development of cropping patterns. It is unfortunately true that the available data base on spatial cropping patterns does not reflect the most recent situation. In order to check whether there have been major shifts in agricultural land use, we inspected the (spatially lumped) temporal

evolution of cropping areas for the different countries and the relevant crops (maize, wheat, sugar beet) based on the FAO statistics (http://www.fao.org/faostat/en/#data; accessed 26 March 2018).

These aggregated data reveal only slight changes in the planting of these major crops over the last 20 years. This supports our assumption that the spatial patterns have not changed much and that our land use data adequately reflect land use for our study period. We will add an explanatory sentence in the main text and add the figure on the temporal trends since 1961 in the SI (see Fig. 1 below).

In this context, we realised that we were not explicit about the crops on which the different herbicides are used. We will add this information in Appendix A4, Table S2.

The second aspect to this comment relates to the crop rotation, which shifts crops between fields for different years. However, these spatial shifts occur at a very local scale and average out at scales relevant for this study. An exception may be the very small catchments (1 – 2 km2) used for model calibration. In these cases however, we rely on specific and detailed land use data for the specific year.

[Figure]

Fig. 1: Temporal evolution of cropping area for four major crops relevant for the modelled herbicide use. The red line indicates the reference year for the spatial cropping patterns. Data source: http://www.fao.org/faostat/en/#data.

P9 L17: Diuron was also used in vineyard still 2008 in France. Is it the same in Germany and Switzerland? Expected impact of missing the agricultural uses in this analysis?

In Switzerland, diuron is indeed still registered and used in vineyards. However, the areal coverage in our study area is of minor importance and therefore neglected.

In Germany, diuron is not registered as agricultural herbicide.

In France, diuron was banned in 2007 with the possibility to use existing stocks by the end of 2008.

In summary, we could safely neglect diuron applications in agriculture for our study area and study period.

P9 L24-29: not clear for me: range of available data? 2008-2012 so 5 years for all the study site excepted for Lorraine? Right? You have to mention the Appendix A5 (figure S3 and S4) to visualize the spatial variability. The reader discovers in the caption of the figure S4 (appendix A5) that the Diuron pattern was just a copy of the carbendazim map. Is it correct? I do not find any discussion on that in the text.

We will refer in the main text to figures S3 and S4 for the spatial patterns.

We will also mention that the spatial diuron and carbendazim patterns are identical because of having the same source areas and uniform application rates across space because there is no data that allows for spatial differentiation.

P9 L34: does any other sources can (even partially) validate this hypothesis?

Unfortunately, we are not aware of any such data source.

P10 L22: "application season" Perhaps explain why to take into account the fact that during application period difference can occur between real applications (unknown) and modelled application (splitted with 1/14 depending of weather windows).

We will add a further explanation.

P10 L22: "error-scaling function" try to better introduce this function, why and how, it's not very clear for the reader.

We will add a further explanation: "The error scaling function makes the standard deviation of herbicide errors proportional to the remaining field stock to reflect that errors are larger in the application period than afterwards, when the compound is present in negligible amounts." This was necessary because the applied Box-Cox transformation alone could not achieve the normality of residuals.

P10 L30: the text and the equation 13 are not supported by any reference

Power transformation of input data and model results is common practice to ensure the normality of residuals. The Box-Cox transformation (Box and Cox 1964) is the most common form of transformation applied for this purpose. We will refer to the iWaQa study (Honti et al. 2017) as a precursor. Equation (13) is a plain normal likelihood function accounting for transformation (Box and Cox 1964).

P11 L10: studies are mentioned to assess the prior distribution of $\varepsilon$ , which one? Wittmer et al (2010) mentioned in L26?

We will rephrase the sentence to " .. from these studies.." such that is clear that we refer to the studies mentioned on p. 11, L . 8 – 9 and listed in Appendix A8, Tab. S4).

P11 L28: I suggest to mention the table S4 and S5 (Appendix A8) after "of the priors for $\varepsilon$ and $\beta$"

We will add this information.

P12 L28: "larger rivers" (Rhine, Aare;" I suggest to mention all main tributaries or use"such as"

We will modify the text accordingly.

P12 L31: The fact that biocide was finally not modelled at the Rhine scale (due to lack of biocides export coefficient in France and Germany) should be precise clearly in the abstract and introduction by differencing the two scales (calibration/validation in the Switzerland scale of herbicides and biocides, and only extrapolation of herbicides at the Rhine scale).

We will make this aspect clear as suggested.

P13 L20: I suggest to link here the table S6 and S7 no called in the main text.

We will add this information as suggested.

P13 L27-28: this sentence needs to be followed by some hypotheses for this bi-modal pattern, especially if physical explanations can help to improve the model for biocides release.

We only can speculate about possible explanations. One possibility we will mention is a bi-modal application pattern. However, we don't have any data on that.

P13 L 32: I can't see how two clusters can be derived from the figure 4. Could you clarify this point?

If the residuals were derived from a single population, the marginal distribution of the (transformed) residuals should follow a unimodal distribution. Inspection of Fig. 4 however (most pronounced TBA, DIU or CBZ, but also visible for the others) reveals that the marginal distribution (for concentrations above base flow concentrations) are bimodal: one modus reveals negative values, one modus positive values.

P13 L39: because the GRI is probably less known compared to the NSE criteria, it could be relevant to provide quality thresholds to consider poor, acceptable, good and very good capabilities of the model.

We understand the wish to have guidelines how to judge any reported GRI value. However, we used this metric as a descriptive tool. Whether or not the reported values are deemed satisfactory or not depends on several factors such as the purpose of the model application, the variables modelled (discharge, nutrients, sediments (see e.g.Moriasi, Arnold et al. 2007), micropollutants etc.), and always has a certain level of subjectivity.

One possibility to overcome (partially) this subjective aspect would be a broad comparison across models and case studies - similar to (Moriasi, Arnold et al. 2007) – for model performance on herbicides and biocides. This would provide an empirical basis of how such models can perform under different conditions. This task however, is beyond the scope of this paper. Therefore, we do not provide any ad-hoc recommendation on what should be considered adequate model performance as measured with the GRI.

P14 L3: could you provide in SI Table S9 the unit of RRMSE (%?) Its not clear in SI if it's really in % (very low value if in %). The table S9 is not used to support quality of the model in the main text.

Probably this comment refers to Table S8 because S9 does not contain RRMSE. The RRMSE is defined in Table 3. It is a unitless quantity. We will indicate this in the caption.

We will refer to Table S9 in the main text.

P14 L2: what could be the interest to reproduce the cumulative concentration (observed and simulated sorted) if the dynamic (timing) is not correctly capture?

For water quality assessment, the exact timing is generally not of interest. The relevant questions are more about the degree, frequency or duration of exceedances of water quality standards. For this reasons, the cumulative distribution is of interest.

We will add a sentence for explanation.

P14 L16: how the reader can derive this information? Which tables or figures supports this statement?

We will add the reference to figures S23 and S24.

P14 L24-25: If I well understand, you observed more dissipation than prior estimation. Does it mean that missing dissipation pathways (leaching to deeper groundwater) could be counterbalance in the model by more sorption (model structure error)?

Actually, we observed lower degradation rates (p. 14, L: 24) but due to higher sorption the mass for transport was available for a longer period.

This observation might counterbalance some model errors, for example regarding the timing of herbicide application. If our modelled input was restricted to a too short period this might be compensated by the parameters characterising the fate in the soil.

We may briefly discuss this possibility in the text.

P15 L2-3: do you mean cumulative distribution or chemogram (dynamic evolution of concentration)?

Yes, we refer to the cumulative distribution and will mention this explicitly in the text.

P15 L4: Could you provide hypothesis or compare this behaviour whith other similar approach (calibration in small scale and validation at large range of scale)?

One hypothesis is that the input estimates are more reliable at the larger scale because regional differences and variabilities in local application dates – hence also input uncertainty - are averaged out. We will mention this in the text.

P15 L15: do you have hypothesis for this over-estimation? Could this overestimation in the validation sites be due to an overestimation of the diuron release in the calibration sites owing to agricultural use neglecting?

Input uncertainty is for sure a very plausible reason. However, agricultural use of diuron is hardly the reason because of the small areal fraction of vineyards.

P15 L26: Could you provide figures or tables (in the main text or SI) to help the reader to understand the differences between the 2 routing methods?

We will add a simple scheme to the SI.

Perhaps it could be relevant to sort the results depending of the catchment area to underline the threshold from which the full routing model improve the concentration prediction.

Sorry, we should have referred to Table S10, where we show for which catchment size routing was considered. We will do so in the revised version.

P15 L39-40: you mentioned previously that you did not simulate the biocide due to a lack of the input database for Germany and France.

We keep this sentence for clarity.

P15 L2: missing figure number! Figure 8, I guessfor

Yes, it should have been Fig. 8.

P15 L4: why applications during fall were not considered? Isoproturon is usually applied in October on winter wheat

There are different reasons for not considering the fall application. First, the fall application was not included in any of the calibration studies. Hence, we lack actual data for model calibration. Second, based on expert interviews with a plant protection specialist in Switzerland, we assumed that most of the isoproturon would be used in spring. Thirdly, predicting the timing of fall application is more complex than the spring application, which can reliably derived from temperature sums during the year. Fall applications also depend on the timing of harvest of the previous crop, which depends on the crops and the growth conditions over the entire growing season. For these reasons, we only considered the spring application.

We discussed this on p. 17, L: 22 – 26. We may add a sentence on how to potentially overcome this shortcoming by deriving an stochastic application model based on application data that may get available from national surveys.

P16 L16: Probably not at the larger scale! It's probably a strong hypothesis for which scenarios with and without river processes (degradation and sorption) could be tested with available dataset (DT50 water...)

As mentioned in the text, this hypothesis (or assumption) is very much compound dependent. Compounds that undergo rapid transformation (e.g., due to hydrolysis, photolysis or biological transformation) had definitely to be treated differently.

However, if a compound is chemically stable, the biological degradation may be not as relevant as possibly expected based on lab-based fate studies. Careful analysis of existing OECD lab data on DT50 reveal that these values have little direct relevance on the actual fate in streams unless the interaction with the sediment layer in the experimental systems and the streams are consistent (Shrestha, Junker et al. 2016).

Some of us have recently analysed longitudinal concentration trends for pharmaceuticals along river Rhine in order to test to which degree degradation and sorption can be derived from such data. It

turns out that first, given the input uncertainty the differentiation between conservative and non-conservative behaviour is hardly possible (Honti, Bischoff et al. submitted).

P16 L15: same comment as previously for unpublished paper (possible in HEES?)

See related response above.

P22 L6: Figure 1 : I suggest a modification of the titre: the study area covered the Rhine viver upstream the Emmerich discharge gauge (red circle) I suggest also to delineate more clearly the Rhine basin with bold line

We will modify the figure as suggested.

P24 L3: Figure 3: I suggest to express concentration in g/L to be able to better link them to EU drinkable threshold (0.1 g/L) I also suggest to add application dates to see interplay between application and rainfall calendars.

We prefer to stick to the ng/L because this is more reader-friendly in figures where low concentrations are depicted (see Fig. S17 – 22).

We will indicate the application periods for the herbicides in the figures.

For biocides, P/Q is not easily understandable, is it a ratio?

Yes, P/Q is the ratio between precipitation and discharge. The reason for this scale is that i) the simulated load is proportional to precipitation (Eq. 8), and that ii) the concentration results from the load divided by discharge.

The sentence (Line 26-27, page 13) that biocide concentration follows rainfall patterns can not easily be derived from this figure), especially at the end because legend hides the rainfall/discharge dynamic (probably high)

We will provide a figure (possibly in the SI) depicting concentrations as a function of the P/Q ratio.

Finally, on the left you have to remove one ng/L (two times in the figure)

We will correct this mistake.

P16 L25: I agree for herbicides, but it could be remembered that seasonal peaks of biocides were not well represented. This weakness seems not to be discussed in the paper. Which types of processes can explain a seasonal variation of biocides exports from roofs and walls?

We mention the limitations regarding biocide concentrations several times in the text and discuss it briefly in 6.2 (Model limitations, p. 17, L: 21, 26 – 27). We add a sentence suggesting that input uncertainty may be the reason behind.

P16 L27: the authors argue that iWaQa can help to identify potential hotspots in river network. I'm not convince according the way that herbicides loads is calculated. Could the authors more clearly underline the strengths and weaknesses of this model to identify hotspot according conceptual structure and processes taken into account? If I well understand, the inputs are derived from

administrative data and can not explain extreme applications and associated peak of concentration at small scale.

We agree that the spatial resolution of "hotspots" in the river network cannot be better than the spatial resolution of herbicide input data. We will clarify this aspect and point out that the model does not aim at identifying single fields in small catchments but at identifying critical regions: "The spatial resolution of such an analysis however, may be strongly limited by a lack of spatial data on compound use and data on local factors influencing transport. Accordingly, it is expected to be valid at a regional instead of a local scale."

P16 L35: missing number of figure Table 5 and Figure : : :?

Sorry for the mistake. It should be Fig. 9.

P17 L5: modify the way that the reference is called. I suggest "as discussed in Honti et al. (2017)

We will modify as suggested.

P17 L21 to27: I suggest to order the different elements (1) too high herbicide: : :, (2) seasonal biocide : : : and (3) the lack of an isoproturon application: : : In the following sentences, I only see discussion on the points 3 and 2 and not for the point 1. Reorganize this section.

We will modify as suggested.

P17 L29: "This agrees with the findings from the error models", I suggest to link here the tables in SI providing range of error.

We will modify as suggested.

P17 L37: missing number for the figure mentioned

Sorry for the mistake. It should be Fig. 10.

P25 Figure 4: even if the calculation is remembered in the caption, it could be relevant to add in the different box of this figure, arrows and captions underlining that if residual > 0 the model underestimated concentration and the opposite if < 0. I also suggest to move this figure in the SI.

We will add this information as suggested and move the figure to the SI as also suggested by Reviewer 2.

P26 L2: I suggest to explain that the catchments are sorted by size (with an arrow and associated surface to better support the sentence (Line 4-5, page 15)

We will modify as suggested.

P28 Figure 7: move to SI

We will add move the figure to the SI as also suggested by Reviewer 2.

P30 Figure 9: move to SI

We will add move the figure to the SI as also suggested by Reviewer 2.

\*\*\*\*\*\*\*\*\*\*\*\*\*\*\*\*\*\*\*\*\*\*\*\*\*\*\*\*\*\*\*\*\*\*\*\*\*\*\*\*

Anonymous Referee #2

Does the paper address relevant scientific questions within the scope of HESS? YES

Thanks.

Does the paper present novel concepts, ideas, tools, or data? YES

Thanks.

Are substantial conclusions reached? YES

Thanks.

Are the scientific methods and assumptions valid and clearly outlined? YES

Thanks.

Are the results sufficient to support the interpretations and conclusions? YES

Thanks.

Is the description of experiments and calculations sufficiently complete and precise to allow their reproduction by fellow scientists (traceability of results)? YES, as far as I can say

Thanks.

Do the authors give proper credit to related work and clearly indicate their own new/original contribution? NO, literature on state-of-the-art is incomplete

We will carefully update the cited references and pay due attention to the suggestions by the reviewer. Thanks for those hints.

Does the title clearly reflect the contents of the paper? YES

Thanks.

Does the abstract provide a concise and complete summary? YES

Thanks.

Is the overall presentation well structured and clear? MOSTLY, some figures are not

We will i) modify some of the figures according to the comments and suggestions by the reviewers (see also response to Reviewer 1).

Is the language fluent and precise? YES

Thanks.

Are mathematical formulae, symbols, abbreviations, and units correctly defined and used? YES

Thanks.

Should any parts of the paper (text, formulae, figures, tables) be clarified, reduced, combined, or eliminated? Substantially clarify figures and/or reduce numbers of figures

See previous comment. We will also move Fig. 4, 7, and 9 to the SI as was also suggested by Reviewer 1.

Are the number and quality of references appropriate? NO (see above)

See response above.

Is the amount and quality of supplementary material appropriate? YES

Thanks.

General Comments:

The authors have used the parsimonious model iWaQa for predicting temporally resolved concentrations of pesticides (from agricultural runoff) and biocides (from diffuse loss from urban areas) in large parts of the Rhine river watershed. The overall goal of the paper was to calibrate the described model and evaluate its prediction capability in a case study in the Rhine basin. I especially appreciate that the authors clearly mention the limitations of the model approach and the time-consuming data collection process often hampered by insufficient data availability.

The manuscript is well-structured and written in good English. Description of the model framework as well as its parameterization and calibration is straightforward and despite its complexity well understandable to me. The line of reasoning in the discussion is mostly convincing to me, but presentation of results is supported by a number of figures and tables that are not always easily understandable and not all are meaningful to me in its current form (see specific comments).

Nevertheless, I can recommend the manuscript to be accepted for publication after minor revision according to the comments below.

Specific comments

Literature with respect to other related model work in this field is not sufficiently reviewed. References are sometimes unfortunate, e.g. MONERIS is cited not with original papers but with a side application and reference to GREAT-ER is outdated (see Kehrein et al., 2015). The EU approach (FOCUS model suite) is neglected as is the DRIPS model (Röpke et al., 2004).

We will carefully update the cited references and pay due attention to the suggestions by the reviewer. Thanks for those hints. Additional references include: (Behrendt, Kornmilch et al. 2002, Röpke, Bach et al. 2004, Kehrein, Berlekamp et al. 2015, Steffens, Jarvis et al. 2015, Villamizar and Brown 2017).

The assumption of general presence of persistent pesticides in baseflow and groundwater (page 6) is somehow terrifying to me, since prevention of elevated concentration levels in groundwater is the main driver for research in this field. Background concentrations determined in the calibration step need to be explicitly compared to the large number of data available. Fortunately, many of these data points are small and below critical limits. These data allow for defining an upper bound for the calibrated background concentrations.

The occurrence of (low) levels of background concentrations of some herbicides in groundwater is a well established fact (Vonberg, Vanderborght et al. 2014, Hakoun, Orban et al. 2017) and supported by our data sets (Gomides Freitas, Singer et al. 2008). An example for this persistence is provided with daily atrazine concentrations in river Rhine at Basel. Even years after the ban of this compound it is ubiquitously found.

[Figure]

Fig. 2: Temporal evolution of atrazine concentrations in river Rhine at Basel. Data source: IRMS.

The background concentration reflects the concentration in the groundwater which is an important contributor of water and substance during base flow conditions. The concentration in the groundwater is assumed to be proportional to the intensity of herbicide use in the area. Hence for simulations the calibrated background concentrations was proportional to areal fraction of the respective crop in a subcatchment. Thus a lower background concentration was introduced at subcatchments with a small share of agricultural area and vice versa a higher background concentration was selected for subcatchments with intensive agriculture land use (as mentioned in Table 1).

The means in the prior distributions of the background concentrations for the calibration were estimated from the values in Table S4.

Many German cities situated in the Rhine catchment use combined sewer systems, whereby a significant fraction of urban surface runoff is directed not instantaneously into rivers (page 6), but to the nearest sewage treatment plant. While the substance-specific load reduction during wastewater treatment is included in the loss rate, it should be discussed whether retardation of transport into the receiving waters for the hydraulic retention time in the sewer system and the wastewater treatment plant could have an effect on the timing of the biocide concentration peaks in the simulations.

The short-term timing is less of an issue for the biocide simulations; the seasonal deviations are more striking. Additionally, the lag time for transport through the urban sewer system causes not a severe problem due to the daily time step used for our simulations. Nevertheless, the point is well taken and we will add a comment in the text.

I think that some figures in the main text are not important enough to justify their presence. Moreover, the meaning of some figures is hardly recognizable from the discussion. I strongly encourage the authors to shift some of the figures to the supporting information. In the main text, I would focus the discussion on the most important points supported by meaningful and easily understandable figures with clearly labelled axes.

See previous comment. We will move Fig. 4, 7, and 9 to the SI as was also suggested by Reviewer 1.

- Figure 4: I could not find a description of the transformation step used to generate the data displayed here. Both axes are displayed in dimensionless scale, but I have no idea what that means. It remains also unclear, what the dashed line represents and why the Figure points to systematic deviations between observed and modelled concentrations (page 13, line 31).
  The reviewer is correct in that the figure was poorly labelled. As described on p. 10, L. 30, the calibration was performed on the Box-Cox-transformed data (Eq. 14, p, 10, L. 36 – p. 11, L: 2). The dashed line represents the regression of the residuals versus the observed concentration (in the transformed space). Without systematic deviations, the regression slope should not deviate from zero.
  We will add the necessary information in the figure and captions. Additionally, it will be moved to the SI. In the main text, we will better explain how to identify the systematic deviations.
- Figure 5: It is not possible to extract the numbers for the GRI values given in the text (page 13/14) from the graph. For me, the range shown in the graph is larger with values up to 5 for the herbicides. Meaning of vertical lines is unclear and the error bars (grey) unreadable. Colour of second section (C-T, C-S) is not identifiable and it is confusing that all categories appear twice on the category axis.
  We will better label the figure and use additional colours to improve its readability. Furthermore, we make the link between statement in the text and the figure more implicit since it seems to be confusing.
- Figure 10: Scaling of the axes is unclear (log-scale of what?)
  The axes represent the average density of data series in space (x-axis) and time (y-axis) across the model domain.
- The x axis represents the number of available data points X across the entire study area (e.g., how many discharge gauging stations, in log scale)

- The y axis gives the number of available data points Y per unit area per decade (, e.g., how many rainfall data for one location in space for a ten year period, in log scale)
- The radius of the circles is proportional to the total number of data points during a unit period of time for the entire study area (product X times Y, in log scale). We will modify the circles such that the circle area is proportional to the log(X Y).

Model performance for biocides is poorer than for pesticides (page 14, line 2), which should be discussed in the light of the different number of calibrated parameters for herbicides (seven/nine with error function) and biocides (just one). Either the biocide model needs more detailed process descriptions or the herbicide model is over-parameterized mutually levelling off uncertainties introduced by the parameters.

The results suggest that the biocide model is overly simplistic because it cannot produce the observed seasonality. However, it is difficult to improve upon this aspect first because the underlying cause is not understood and second because of poor input data that preclude a more reliable simulation. We will add a sentence in the discussion to elaborate more on this issue beyond we already have on p. 17, L. 34 – 38.

Minor comments

- pathways, wastewater (page 2)
  Will be corrected.
- … where M(t) [g d-1] is the mass ratio … (page 5)
  We will reword to " is the rate of mass applied…"
- … temperature sum models is used (page 5)
  Will be corrected.
- … of the pronounced concentrations peaks (page 15)
  Will be corrected.
- Number of Figure missing (page 16, line 2)
  The correct figure number will be inserted.
- … for the model compounds considered in this paper (page 16)
  Will be corrected.
- …micropollutants from points sources (page 16)
  Will be corrected.
- …purely hydrological models can accomplish (page 16)
  Will be corrected.
- Number of Figure missing (page 17, last but one line)
  The correct figure number will be inserted.
* * *
Anonymous Referee #3

The authors present an extensive modeling study on pesticide transport in the Rhine basin. The paper including the supplement is very long (as is the river Rhine). It builds on quite an impressive amount of work and is well written, although not everything is explained in full detail. The latter is probably unavoidable with such kind of studies. I appreciate that the authors provide there input data and model code.

The basic idea is to set up a simple, parsimonious model, to calibrate it with detailed data (from Switzerland, where a number of very good, detailed studies were carried out in the past) and predict transport on large scale. The approach is actually quite successful. Modeled concentrations partly deviate from measured concentrations by up to a factor of 5 or so, but we should also not forget that the concentrations are low.

It is clear that the model can be improved when better input data become available (if ever).

I have only one major point to make. I refer to the last paragraph of section 5.1. The authors used a (lumped) modeling approach in which sorbed phase and solution phase concentration are expressed in total masses in the system (watershed). The model was parameterized with laboratory data from a study by Freitas et al. 2008 (reference missing), but it is not explained how.

Was the different soil to solution ratio in the lab considered? If the model is directly applied to the lab data ($M_s$=sorbed, $M_w$=dissolved), it would yield too low partitioning coefficients, what is indeed what the authors observed (section 5.1, last paragraph). This is so because, as a physicochemical fact, the sorption isotherm is independent of the soil to solution ratio as has been shown many times in the literature. In the lab, the mass in solution is much higher (by a factor of 5 or so, depending on the soil to solution ratio used). Taking it differently, sorption isotherms measured in the lab cannot be applied to a system (watershed) without making an assumption about the soil to solution ratio in it.

The reference was included in the reference list but as Gomides Freitas et al., 2008. We will correct this mistake and make the referencing consistent.

The data stem from a field study (p. 11, L. 29 – 30). Pore water concentrations from soil samples were obtained upon extraction of 20 g of soil with 40 mL 0.01 M CaCl2 solution. Therefore, the reviewer is right in that the soil to solution ratio does not correspond to the conditions in field soils. How this fact affects the partitioning between the aqueous and the solid phase depends on the sorption isotherm. With a linear isotherm, the distribution coefficient Kd is constant irrespective of the soil-solution ratio. With non-linear isotherms this is not the case. Our field data indicate non-linear sorption behaviour that can described with a Freundlich isotherm.

Based on the experimental data, we can calculate how the (apparent) distribution coefficient Kd is expected to change upon a modification of the soil-solution ratio. To this end, we used the S-metolachlor data from (Gomides Freitas 2005) with a Freundlich exponent n of 0.7 and a Freundlich coefficient of 17 and calculated the distribution coefficient between solid and the aqueous phase for the lab conditions (see above, rho = 0.33, theta = 0.67) and for realistic field conditions (rho = 1.00, theta = 0.25). The resulting ratios between the lab and field distribution coefficients are depicted in Fig. 3 across a wide range of total concentrations (10 to 2500 microgram/kg soil).

These calculations first demonstrate that the Reviewer is correct in that the lab data underestimate sorption. Second, the data also reveal that the degree of underestimation is rather limited such that it won't affect our findings substantially. Nevertheless, we will add a comment stating that the different soil-solution ratios between the measuring conditions and the field situation will have contributed to the underestimation of partitioning.

[Figure]

Fig. 3: Ratio between distribution coefficients in the field and in the lab for S-metolachlor based on the data reported in (Gomides Freitas 2005). The red line indicates identity between the lab and the field data.

Unfortunately, my print-out version did not contain page numbers.

page 1, line 17 to what degree

Will be corrected.

At some locations in the text, references to figures lack the number (figure [empty]).

Will be corrected (see response above).

eq 1: $M(t)$ is undefined (cf. eq 8)

This mistake will be corrected.

eq 8: in eq 2 and 3, $M_a$ was defined as the mass applied to the catchment the mass present in the catchment should be $M$ as in the LHS of eq 1

Will be corrected.

Figure 4: how were the residuals transformed?

All data are Box-Cox transformed, see response to Reviewer 1 on the same issue.

Figure 5 Overview of

Will be corrected.

Figure 6: I did not get the rational behind the fold difference. This should be explained in the text and in the figure caption.

In the caption, we will refer to the description in Tab. 3 and we will add additional explanations in the text saying that this metric allows to quantify the deviation between observations and simulations irrespective of whether observations were under- or over estimated.

Figure 7: What is the violin about? Is the envelope a standard error? Explain in the text and in the figure caption.

Violin plots are a combination of a boxplot and a kernel density plot. We will add a further explanation in text and captions and refer to the original publication introducing this kind of plot (Hintze and Nelson 1998).

Figure 9: Explain the envelope?

Violin plots are a combination of a boxplot and a kernel density plot. We will add a further explanation in text and captions and refer to the original publication introducing this kind of plot (Hintze and Nelson 1998).

Figure 10: I have no clue what this figure is about. Explain the axes, assign units. Explain properly or delete the whole thing.

We admit that the figure was poorly explained. We will add the information provided in the corresponding response to Reviewer 2 (see above).

Table 1: Did you properly explain the scaling factor ? I think no.

The scaling factor is introduced and explained on p. 10, L. 20 – 26. In accordance with the reply to reviewer #1 on the same issue, we will add further explanation

Table 2: Explain the abbreviations (like NADUF).

We will add these explanation (NADUF: NADUF – National long-term surveillance of Swiss rivers, IRMS: International Rhine Monitoring Station, NAWA SPEZ: National Surface Water Quality Monitoring Programme) and make sure they are used in consistent forms across the manuscript.

Table 3: Explain the symbols in the caption.

We explain the symbols

Table 4: Explain abbreviations.

We will add these explanations.

Table 5: C and V should be defined. Atrazine

We will explain these symbols (C = calibration, V = validation). It is unclear what the reviewer wanted to say about atrazine.

***Summing up, the tables and figures should be self-explaining.

We agree and improve according to the responses above.

References:

Behrendt, H., M. Kornmilch, D. Opitz, O. Schmoll and G. Scholz (2002). "Estimation of the nutrient inputs into river systems - experiences from German rivers." Journal of Material Cycles and Waste Management **3**(1-3): 107-117.

Box, G. E. P. and D. R. Cox (1964). "An analysis of transformations." Journal of the Royal Statistical Society, Series B. **26** 211–252.

Gomides Freitas, L. (2005). Herbicide losses to surface waters in a small agricultural catchment Diss ETH 16076, Swiss Federal Institute of Technology.

Gomides Freitas, L., H. Singer, S. R. Müller, R. Schwarzenbach and C. Stamm (2008). "Source area effects on herbicide losses to surface waters - A case study in the Swiss Plateau." Agriculture, Ecosystems & Environment **128**: 177 - 184, doi:110.1016/j.agee.2008.1006.1014.

Hakoun, V., P. Orban, A. Dassargues and S. Brouyère (2017). "Factors controlling spatial and temporal patterns of multiple pesticide compounds in groundwater (Hesbaye chalk aquifer, Belgium)." Environmental Pollution **223**: 185-199.

Hintze, J. L. and R. D. Nelson (1998). "Violin Plots: A box plot-density trace synergism." The American Statistician **52**: 181-184.

Hirsch, R. M. (1979). "An evaluation  of some record reconstruction techniques " Water Resouces Research **15**: 1781 - 1790.

Honti, M., F. Bischoff, A. Moser, C. Stamm and K. Fenner (submitted). "Micropollutant degradation in rivers: Suitability of field monitoring and regulatory data."

Kehrein, N., J. Berlekamp and J. Klasmeier (2015). "Modeling the fate of down-the-drain chemicals in whole watersheds: New version of the GREAT-ER software." Environmental Modelling & Software **64**: 1-8.

Moriasi, D. N., J. G. Arnold, M. W. Van Liew, R. L. Bingner, R. D. Harmel and T. L. Veith (2007). "Model evaluation guidelines for systematic quantification of accurracy in watershed simulations." Transactions of the American Society of Agricultural and Biological Engineers **50**: 885 - 900.

Röpke, B., M. Bach and H. G. Frede (2004). "DRIPS - A DSS for estimating the input quantity of pesticides for German river basins." Environmental Modelling and Software **19**(11): 1021-1028.

Shrestha, P., T. Junker, K. Fenner, S. Hahn, M. Honti, R. Bakkour, C. Diaz and D. Hennecke (2016). "Simulation studies to explore biodegradation in water-sediment systems: from OECD 308 to OECD 309. ." Environmental Science and Technology **50**: 6856-6864.

Steffens, K., N. Jarvis, E. Lewan, B. Lindström, J. Kreuger, E. Kjellström and J. Moeys (2015). "Direct and indirect effects of climate change on herbicide leaching - A regional scale assessment in Sweden." Science of the Total Environment **514**: 239-249.

Villamizar, M. L. and C. D. Brown (2017). "A modelling framework to simulate river flow and pesticide loss via preferential flow at the catchment scale." Catena **149**: 120 - 130.

Vonberg, D., J. Vanderborght, N. Cremer, T. Pütz, M. Herbst and H. Vereecken (2014). "20 years of long-term atrazine monitoring in a shallow aquifer in western Germany." Water Research **50**: 294-306.

---

## Author Response (AR1)

**Revision of the manuscript (point by point explanations)**

To all reviewers

Thanks for the detailed and constructive comments. Below we added our initial responses (in red) to each single point and listed how the manuscript has finally be modified (in **green, line numbers refer to the revised version**).

We also appended at the end the revised manuscript in the track-change mode such that one can easily identify where the paper was revised.

On behalf of all co-authors.

Christian Stamm

Anonymous Referee #1

General comments

*Developping and testing the limits of a parsimonious model of micropollutants transport from catchment to river at various scales is relevant both for stakeholders and for the scientific community to address the degree of simplification required/able to capture the micropollutants patterns. The key concept of this approach (link the load of micropollutants to the discharge and/or the rain) is not new but the approach to validate at small scale with European databases the load then upscaling the approach at larger scale reaching the main part of the Rhine catchment is new. The spatial preprocessing of existing European data to improve the information for the subcatchment can also be underlined. The state of the art on the different components covered by the work is well presented with relevant references. The model development, main hypotheses and calibration/validation/transposition steps are clearly presented.*

Thanks for the positive feedback.

*However, a scheme summarizing spatial and temporal discretization with associated processes across scale (calibration then validation catchments and full Rhine scale) is clearly missing in the main text. A very simple scheme (focused on subcatchment delineation) is presented in SI but can't play this summary goal.*

We agree that such a scheme would be useful and will include an improved version of the current figure in the SI into the main text.

**We have first modified Fig. 2 such as to clearly show the spatial arrangement of the calibration and validation catchments within the river network. Additionally, we have added a schematic representation in the SI that depicts also the size relations between the catchments (Fig. S4).**

*The results are clearer for herbicides than for biocides. The reader discovered different hypotheses that reduce progressively the extent of the biocides loads ant transport modelling at larger scale. I wonder if a focus on herbicides only, should not be better and stronger.*

We agree that the data basis for modelling biocides is substantially poorer than for herbicides. However, we consider biocides as relevant compounds and find it thus important to report the modelling challenges and thereby illustrate the limitations when simulating such compounds.

**No changes were made.**

*I detailed below specific comments and corrections*

*P1 L30: check homogeneity (S-metolachlor and metolachlor are used in the text)*

Thanks for the careful reading. We will check for consistency, use consistent terminology and make it explicit if one has to switch to the other expression because data do not allow the specific use of S-metolachlor.

**We corrected the terminology throughout the manuscript incl. the SI.**

*P2 L28: Missing dot at the end of a sentence "2013) One"*

Will be corrected.

**Corrected.**

*P3 L13-14: original aspect of this work +add that it's a daily or hourly time step*

We will add the temporal resolution.

**We have modified the text to i) highlight the novelty more and to be explicit about the temporal resolution.**

**"Here we present a model that covers major urban and agricultural sources for pesticides in streams that can be applied to large water basins, provides high spatial and temporal resolution (hourly to daily) and is still parsimonious. It is similar to the iWaQa model approach in (Honti et al., 2017) but adapted for large basins by including an explicit routing component by coupling it to the AQUASIM model. It differs from many other model concepts in that it does not include a rainfall-runoff module but directly links agricultural pesticide losses in a novel way to measured discharge and urban biocide losses directly to precipitation." (L. 94 - 99).**

*P3 L23: all the basin? just after it's mentioned that only the basin upstream of the station Emmerich am Rhein is covered by this study, precise*

We will mention here that the basin upstream of Emmerich is covered.

**Done.**

*P4 L6: general comment: An overall scheme of the model should be relevant to improve understanding of the spatial links between objects, considered processes and links between AWaQa and AQUASIM (what's happened in the sub-catchment scale and then in the river, degradation, … ) The appendix A2 is one aspect of the discussion but it not covers the processes.*

As mentioned above, we agree and will add a corresponding scheme.

**We added two additional figures to the Appendix that show first how output from the substance transfer module from each catchment is routed with the two routing approaches and second the spatial arrangement of the study catchments (Fig. S3, S8).**

*P5 L8 and L9: in the equations 2 and 3 Kdeg has to be used instead of Kd to be homogeneous with the rest of the paper*

Thanks for noticing. This will be corrected.

**Corrected.**

*P5 L11: not clear at this stage how the available fraction is link to rainfall*

This fraction is not linked to rainfall but describes the effect of immediate sorption in contrast to the slower exchange processes described by Eq. [3]. We try to rephrase such to avoid misunderstandings.

Suggested sentence: "… available for transport, such that it can be directly mobilised when it rains."

**We added the sentence as mentioned above. (L. 175)**

*P5 L21: interesting but why 1/14? Expert panel, reasonable fractionation? Sensitivity of the model to this fraction?* Does it means that 14 days are required to consider 100% of application? Can this hypothesis impact peak modelling due to dilution of input signal?

Yes, the total application mass is distributed over 14 days. This reflects the fact that pesticides are not applied on all fields in a subcatchment on the same day. With applications taking place only during dry days, the total application period lasts 3 – 4 weeks which is reasonable based on our field experience.

These assumptions have of course an influence on simulated peak concentrations because they only reflect an approximation to the actual temporal application pattern, which is unfortunately not known.

**No changes were made.**

P6 L23: no rainfall dependency?

Yes, the load is proportional to rainfall - the last term (P(t)) represents daily precipitation.

**No changes were made.**

P6 L34-35: not clear for me, does it mean that any transformation is considered in the river routing model?

Yes, the load aggregation models neglects transformation processes while the routing procedure with Aquasim in principle could.

**No changes were made.**

P7 L8: should be interesting to mention in the abstract and in the introduction...

We will mention the coupling with Aquasim in both parts.

**Done (L. 28, 97).**

P7 L11-12: is it possible to mention an unpublished paper in HESS?

We will skip this reference and the one on page 16

**The reference was skipped.**

P7 L17: I think it is the table S2 (in the appendix A4) and not the table S4.

Thanks for noticing. This will be corrected.

**Corrected.**

P7 L20: not easy to understand even if it's describe below... a scheme should be relevant to improve understanding.

We will add a schematic representation for better illustration.

**Done. See explanation above (Fig. S3, S8).**

P7 L25: what % considered considering strahler order less than five?

We will add the respective information.

**We added "(804 out 931 or 86 % of the available gauging stations)." (L. 268)**

P7 27: Explain briefly the area ratio method to help the reader at this stage

The area ratio method simply assumes that discharge scales proportional to catchment area (see e.g. Hirsch 1979). We will add this explanation to the sentence: "…, which assumes that discharge scales proportional to catchment area." .

**We added the explanation as mentioned above. (L. 270 – 271)**

P7 31: Same comment, explain briefly the map-correlation method to help the reader at this stage.

We will add a sentence for more explicit explanation: "Selection of the reference stations is based on the map-correlation method from Archfield and Vogel. (2010). This geostatistical method calculates the correlation between discharge time series at observed stream gauges and estimates the station with the most correlated discharge at the ungauged catchment based."

**We added the explanation as mentioned above. (L. 274 – 277).**

P8 L16-18: not clear for me! –6.5 per km (in z?)

We will rephrase ("From these altitude deviations, temperature values were corrected based on a temperature decrease by -0.0065°C/m altitude increase".)

**Done. (L. 301 – 302)**

P8 L21: Expected evolution if 2000 is the reference? Impact of rotation of crops (spatial difference between years)?

This is a very important point and there are two aspects to this question. The first relates to the long-term development of cropping patterns. It is unfortunately true that the available data base on spatial cropping patterns does not reflect the most recent situation. In order to check whether there have been major shifts in agricultural land use, we inspected the (spatially lumped) temporal evolution of cropping areas for the different countries and the relevant crops (maize, wheat, sugar beet) based on the FAO statistics (http://www.fao.org/faostat/en/#data; accessed 26 March 2018).

These aggregated data reveal only slight changes in the planting of these major crops over the last 20 years. This supports our assumption that the spatial patterns have not changed much and that our land use data adequately reflect land use for our study period. We will add an explanatory sentence in the main text and add the figure on the temporal trends since 1961 in the SI (see Fig. 1 below).

**We added the following text:**

**"Because there was no data set available reflecting the most recent situation, we checked whether there have been major shifts in agricultural land use with the spatially lumped data on the temporal evolution of cropping areas for the different countries and the relevant crops (maize, wheat, sugar beet) based on the FAO statistics (http://www.fao.org/faostat/en/#data; accessed 26 March 2018). These aggregated data reveal mostly little changes in the planting of these major crops over the last 20 years. This supports our assumption that the spatial patterns have not changed much and that our land use data adequately reflect land use for our study period (see Fig. S4). For Switzerland, more recent land use (2004 – 2009) and crop statistics (2010) were available and used." (L. 308 – 314)**

In this context, we realised that we were not explicit about the crops on which the different herbicides are used. We will add this information in Appendix A4, Table S2.

**We added this information in Appendix A4, Table S1.**

The second aspect to this comment relates to the crop rotation, which shifts crops between fields for different years. However, these spatial shifts occur at a very local scale and average out at scales relevant for this study. An exception may be the very small catchments (1 – 2 km2) used for model calibration. In these cases however, we rely on specific and detailed land use data for the specific year.

**No changes were made with regard to crop rotations**

[Figure]

Fig. 1: Temporal evolution of cropping area for four major crops relevant for the modelled herbicide use. The red line indicates the reference year for the spatial cropping patterns. Data source: http://www.fao.org/faostat/en/#data.

P9 L17: Diuron was also used in vineyard still 2008 in France. Is it the same in Germany and Switzerland? Expected impact of missing the agricultural uses in this analysis?

In Switzerland, diuron is indeed still registered and used in vineyards. However, the areal coverage in our study area is of minor importance and therefore neglected.

In Germany, diuron is not registered as agricultural herbicide.

In France, diuron was banned in 2007 with the possibility to use existing stocks by the end of 2008.

In summary, we could safely neglect diuron applications in agriculture for our study area and study period.

**No changes were made.**

P9 L24-29: not clear for me: range of available data? 2008-2012 so 5 years for all the study site excepted for Lorraine? Right? You have to mention the Appendix A5 (figure S3 and S4) to visualize the spatial variability. The reader discovers in the caption of the figure S4 (appendix A5) that the Diuron pattern was just a copy of the carbendazim map. Is it correct? I do not find any discussion on that in the text.

We will refer in the main text to figures S3 and S4 for the spatial patterns.

**We added the sentence "The resulting spatial distribution of estimated input is depicted in Appendix A6, Fig. S5."**

We will also mention that the spatial diuron and carbendazim patterns are identical because of having the same source areas and uniform application rates across space because there is no data that allows for spatial differentiation.

**We added "Because of the lack of spatially distributed biocide use data, the spatial distributions of CBZ and DIU are identical (see Appendix A6, Fig. S6). (L. 376 – 377)**

P9 L34: does any other sources can (even partially) validate this hypothesis?

Unfortunately, we are not aware of any such data source.

**No changes were made.**

P10 L22: "application season" Perhaps explain why to take into account the fact that during application period difference can occur between real applications (unknown) and modelled application (splitted with 1/14 depending of weather windows).

We will add a further explanation.

**We think that the explanation on the error-scaling function provided below is sufficient at that point.**

P10 L22: "error-scaling function" try to better introduce this function, why and how, it's not very clear for the reader.

We will add a further explanation: "The error scaling function makes the standard deviation of herbicide errors proportional to the remaining field stock to reflect that errors are larger in the application period than afterwards, when the compound is present in negligible amounts." This was necessary because the applied Box-Cox transformation alone could not achieve the normality of residuals.

**We added the sentence as described above. L. 396 – 398.**

P10 L30: the text and the equation 13 are not supported by any reference

Power transformation of input data and model results is common practice to ensure the normality of residuals. The Box-Cox transformation (Box and Cox 1964) is the most common form of transformation applied for this purpose. We will refer to the iWaQa study (Honti et al. 2017) as a precursor. Equation (13) is a plain normal likelihood function accounting for transformation (Box and Cox 1964).

**We modified the sentence: "The likelihood function used in this study is based on the assumption that Box-Cox transformed (Box and Cox, 1964) time series of concentration data C lead to independent and identically distributed normal errors as described in Honti et al., (2017). The corresponding likelihood function is as follows:…." . (L. 405 – 407)**

P11 L10: studies are mentioned to assess the prior distribution of $\varepsilon$ , which one? Wittmer et al (2010) mentioned in L26?

We will rephrase the sentence to " .. from these studies.." such that is clear that we refer to the studies mentioned on p. 11, L . 8 – 9 and listed in Appendix A8, Tab. S4).

**Done. L. 424, Tab. S3.**

P11 L28: I suggest to mention the table S4 and S5 (Appendix A8) after "of the priors for $\varepsilon$ and $\beta$"

We will add this information.

**Done. L. 442.**

P12 L28: "larger rivers" (Rhine, Aare;" I suggest to mention all main tributaries or use"such as"

We will modify the text accordingly.

**Done. L. 476.**

P12 L31: The fact that biocide was finally not modelled at the Rhine scale (due to lack of biocides export coefficient in France and Germany) should be precise clearly in the abstract and introduction by differencing the two scales (calibration/validation in the Switzerland scale of herbicides and biocides, and only extrapolation of herbicides at the Rhine scale).

We will make this aspect clear as suggested.

**We modified the respective sentences to:**

**"Subsequently, it was validated for herbicides and biocides in Switzerland for different years on 12 catchments of much larger size (31 –36'000 km$^2$) and for herbicides for the entire Rhine basin upstream of the Dutch-German border (160'000 km$^2$) without any modification." L. 31 – 34.**

**In the Introduction we modified the last sentence: "Due to lack of data, the biocide part was only tested within Switzerland." L. 109 – 110.**

P13 L20: I suggest to link here the table S6 and S7 no called in the main text.

We will add this information as suggested.

**Done. L. 505.**

P13 L27-28: this sentence needs to be followed by some hypotheses for this bi-modal pattern, especially if physical explanations can help to improve the model for biocides release.

We only can speculate about possible explanations. One possibility we will mention is a bi-modal application pattern. However, we don't have any data on that.

**We added the following sentence:**

**"A possible reason for this temporal pattern is a seasonal application pattern of biocide application. However, there is no data available for testing this hypothesis." L. 518 – 519.**

P13 L 32: I can't see how two clusters can be derived from the figure 4. Could you clarify this point?

If the residuals were derived from a single population, the marginal distribution of the (transformed) residuals should follow a unimodal distribution. Inspection of Fig. 4 however (most pronounced TBA, DIU or CBZ, but also visible for the others) reveals that the marginal distribution (for concentrations above base flow concentrations) are bimodal: one modus reveals negative values, one modus positive values.

**Fig. 4 was moved to the SI (new: Fig. S11). No further changes were made.**

P13 L39: because the GRI is probably less known compared to the NSE criteria, it could be relevant to provide quality thresholds to consider poor, acceptable, good and very good capabilities of the model.

We understand the wish to have guidelines how to judge any reported GRI value. However, we used this metric as a descriptive tool. Whether or not the reported values are deemed satisfactory or not depends on several factors such as the purpose of the model application, the variables modelled (discharge, nutrients, sediments (see e.g.Moriasi, Arnold et al. 2007), micropollutants etc.), and always has a certain level of subjectivity.

One possibility to overcome (partially) this subjective aspect would be a broad comparison across models and case studies - similar to (Moriasi, Arnold et al. 2007) – for model performance on herbicides and biocides. This would provide an empirical basis of how such models can perform under different conditions. This task however, is beyond the scope of this paper. Therefore, we do not provide any ad-hoc recommendation on what should be considered adequate model performance as measured with the GRI.

**No changes were made.**

P14 L3: could you provide in SI Table S9 the unit of RRMSE (%?) Its not clear in SI if it's really in % (very low value if in %). The table S9 is not used to support quality of the model in the main text.

Probably this comment refers to Table S8 because S9 does not contain RRMSE. The RRMSE is defined in Table 3. It is a unitless quantity. We will indicate this in the caption.

**Done.**

We will refer to Table S9 in the main text.

**Done. L. 509.**

P14 L2: what could be the interest to reproduce the cumulative concentration (observed and simulated sorted) if the dynamic (timing) is not correctly capture?

For water quality assessment, the exact timing is generally not of interest. The relevant questions are more about the degree, frequency or duration of exceedances of water quality standards. For this reasons, the cumulative distribution is of interest.

We will add a sentence for explanation.

**We modified the sentence by inserting:**

**– this generally relevant for water quality assessment -  (L. 531)**

P14 L16: how the reader can derive this information? Which tables or figures supports this statement?

We will add the reference to figures S23 and S24.

**These figures are mentioned in the text. Now Fig. S27 – S28, see L. 548.**

P14 L24-25: If I well understand, you observed more dissipation than prior estimation. Does it mean that missing dissipation pathways (leaching to deeper groundwater) could be counterbalance in the model by more sorption (model structure error)?

Actually, we observed lower degradation rates (p. 14, L: 24) but due to higher sorption the mass for transport was available for a longer period.

This observation might counterbalance some model errors, for example regarding the timing of herbicide application. If our modelled input was restricted to a too short period this might be compensated by the parameters characterising the fate in the soil.

We may briefly discuss this possibility in the text.

**We added a sentence:**

**"However, stronger sorption in the model could also compensate for pesticide applications that were missed by the model." L. 556 – 557.**

P15 L2-3: do you mean cumulative distribution or chemogram (dynamic evolution of concentration)?

Yes, we refer to the cumulative distribution and will mention this explicitly in the text.

**We inserted "(cumulative)" L. 576.**

P15 L4: Could you provide hypothesis or compare this behaviour whith other similar approach (calibration in small scale and validation at large range of scale)?

One hypothesis is that the input estimates are more reliable at the larger scale because regional differences and variabilities in local application dates – hence also input uncertainty - are averaged out. We will mention this in the text.

**We added a sentence: "This might be explained by averaging out regional differences and variabilities in local application dates – hence also input uncertainty - across larger scales. L. 579 – 580.**

P15 L15: do you have hypothesis for this over-estimation? Could this overestimation in the validation sites be due to an overestimation of the diuron release in the calibration sites owing to agricultural use neglecting?

Input uncertainty is for sure a very plausible reason. However, agricultural use of diuron is hardly the reason because of the small areal fraction of vineyards.

**No changes were made.**

P15 L26: Could you provide figures or tables (in the main text or SI) to help the reader to understand the differences between the 2 routing methods?

We will add a simple scheme to the SI.

**We added Fig. S3 to that end.**

Perhaps it could be relevant to sort the results depending of the catchment area to underline the threshold from which the full routing model improve the concentration prediction.

Sorry, we should have referred to Table S10, where we show for which catchment size routing was considered. We will do so in the revised version.

**Done. L. 603.**

P15 L39-40: you mentioned previously that you did not simulate the biocide due to a lack of the input database for Germany and France.

We keep this sentence for clarity.

**No changes were made.**

P15 L2: missing figure number! Figure 8, I guessfor

Yes, it should have been Fig. 8.

**Done. L. 586.**

P15 L4: why applications during fall were not considered? Isoproturon is usually applied in October on winter wheat

There are different reasons for not considering the fall application. First, the fall application was not included in any of the calibration studies. Hence, we lack actual data for model calibration. Second, based on expert interviews with a plant protection specialist in Switzerland, we assumed that most of

the isoproturon would be used in spring. Thirdly, predicting the timing of fall application is more complex than the spring application, which can reliably derived from temperature sums during the year. Fall applications also depend on the timing of harvest of the previous crop, which depends on the crops and the growth conditions over the entire growing season. For these reasons, we only considered the spring application.

We discussed this on p. 17, L: 22 – 26. We may add a sentence on how to potentially overcome this shortcoming by deriving an stochastic application model based on application data that may get available from national surveys.

**We added a sentence: "In the future, this deficit may be overcome by deriving a stochastic application model based on application data obtained from national surveys." L. 688.**

P16 L16: Probably not at the larger scale! It's probably a strong hypothesis for which scenarios with and without river processes (degradation and sorption) could be tested with available dataset (DT50 water...)

As mentioned in the text, this hypothesis (or assumption) is very much compound dependent. Compounds that undergo rapid transformation (e.g., due to hydrolysis, photolysis or biological transformation) had definitely to be treated differently.

However, if a compound is chemically stable, the biological degradation may be not as relevant as possibly expected based on lab-based fate studies. Careful analysis of existing OECD lab data on DT50 reveal that these values have little direct relevance on the actual fate in streams unless the interaction with the sediment layer in the experimental systems and the streams are consistent (Shrestha, Junker et al. 2016).

Some of us have recently analysed longitudinal concentration trends for pharmaceuticals along river Rhine in order to test to which degree degradation and sorption can be derived from such data. It turns out that first, given the input uncertainty the differentiation between conservative and non-conservative behaviour is hardly possible (Honti, Bischoff et al. submitted).

**No changes were made.**

P16 L15: same comment as previously for unpublished paper (possible in HEES?)

See related response above.

**Done.**

P22 L6: Figure 1 : I suggest a modification of the titre: the study area covered the Rhine viver upstream the Emmerich discharge gauge (red circle) I suggest also to delineate more clearly the Rhine basin with bold line

We will modify the figure as suggested.

**We modified Fig. 1 as suggested.**

**Title: Because the title does not imply that the simulations covered the entire basin, we suggest to keep the title for the sake of brevity.**

P24 L3: Figure 3: I suggest to express concentration in g/L to be able to better link them to EU drinkable threshold (0.1 g/L) I also suggest to add application dates to see interplay between application and rainfall calendars.

We prefer to stick to the ng/L because this is more reader-friendly in figures where low concentrations are depicted (see Fig. S17 – 22).

We will indicate the application periods for the herbicides in the figures.

**We indicated the application periods for the three herbicides in Fig. 3.**

For biocides, P/Q is not easily understandable, is it a ratio?

Yes, P/Q is the ratio between precipitation and discharge. The reason for this scale is that i) the simulated load is proportional to precipitation (Eq. 8), and that ii) the concentration results from the load divided by discharge.

**No changes were made.**

The sentence (Line 26-27, page 13) that biocide concentration follows rainfall patterns can not easily be derived from this figure), especially at the end because legend hides the rainfall/discharge dynamic (probably high)

We will provide a figure (possibly in the SI) depicting concentrations as a function of the P/Q ratio.

**We modified Fig. 3 such that the high values at the end of the year can be properly seen. Additionally, we added Fig. S10.**

Finally, on the left you have to remove one ng/L (two times in the figure)

We will correct this mistake.

**Done.**

P16 L25: I agree for herbicides, but it could be remembered that seasonal peaks of biocides were not well represented. This weakness seems not to be discussed in the paper. Which types of processes can explain a seasonal variation of biocides exports from roofs and walls?

We mention the limitations regarding biocide concentrations several times in the text and discuss it briefly in 6.2 (Model limitations, p. 17, L: 21, 26 – 27). We add a sentence suggesting that input uncertainty may be the reason behind.

**We added "… while that of biocides was not well reflected (see below)" L. 641, and referred to the limited model performance for biocides at several locations (L. 532, 575, 659, 689 – 692)**

P16 L27: the authors argue that iWaQa can help to identify potential hotspots in river network. I'm not convince according the way that herbicides loads is calculated. Could the authors more clearly underline the strengths and weaknesses of this model to identify hotspot according conceptual structure and processes taken into account? If I well understand, the inputs are derived from administrative data and can not explain extreme applications and associated peak of concentration at small scale.

We agree that the spatial resolution of "hotspots" in the river network cannot be better than the spatial resolution of herbicide input data. We will clarify this aspect and point out that the model does not aim at identifying single fields in small catchments but at identifying critical regions: "The spatial resolution of such an analysis however, may be strongly limited by a lack of spatial data on compound use and data on local factors influencing transport. Accordingly, it is expected to be valid at a regional instead of a local scale."

**We added the explanation as mentioned above. L. 642 - 644.**

P16 L35: missing number of figure Table 5 and Figure : : :?

Sorry for the mistake. It should be Fig. 9.

**Done. Now Fig. S34, see L. 653.**

P17 L5: modify the way that the reference is called. I suggest "as discussed in Honti et al. (2017)

We will modify as suggested.

**Done. L. 663.**

P17 L21 to27: I suggest to order the different elements (1) too high herbicide: : :, (2) seasonal biocide : : : and (3) the lack of an isoproturon application: : : In the following sentences, I only see discussion on the points 3 and 2 and not for the point 1. Reorganize this section.

We will modify as suggested.

**We have extended this paragraph to the following version:**

**"The first problem of herbicide background concentration levels would require a more explicit modelling of the long-term fate of these compounds in the coupled unsaturated and saturated zone. To keep such a model parsimonious one had to test whether these background concentrations could be empirically linked to some simple catchment characteristics. Second, the herbicide application in fall for example is much more difficult to predict compared to the spring application because it not only depends on a single variable such as the temperature sum over the year but it is also influenced by all the climatic variables determining the time of cropping of the previous crop and potential intercropping. In the future, this deficit may be overcome by deriving an stochastic application model based on application data obtained from available from national surveys. Regarding the third aspect, the seasonal biocide patterns, we lack any information about biocide use on buildings that could explain the observed seasonality. Targeted surveys on actual use across the year might be a solution." L. 682 – 692.**

P17 L29: "This agrees with the findings from the error models", I suggest to link here the tables in SI providing range of error.

We will modify as suggested.

**We refer to Appendix, Fig. S27, S28, L. 694. .**

P17 L37: missing number for the figure mentioned

Sorry for the mistake. It should be Fig. 10.

**Corrected. Now Fig. 7, L. 702.**

P25 Figure 4: even if the calculation is remembered in the caption, it could be relevant to add in the different box of this figure, arrows and captions underlining that if residual > 0 the model underestimated concentration and the opposite if < 0. I also suggest to move this figure in the SI.

We will add this information as suggested and move the figure to the SI as also suggested by Reviewer 2.

**Done. See Fig. S11.**

P26 L2: I suggest to explain that the catchments are sorted by size (with an arrow and associated surface to better support the sentence (Line 4-5, page 15)

We will modify as suggested.

**We indicated now the size of the catchments but didn't add an arrow to not overload the figure. (Fig. 4).**

P28 Figure 7: move to SI

We will add move the figure to the SI as also suggested by Reviewer 2.

**Done. Now Fig. S33.**

P30 Figure 9: move to SI

We will add move the figure to the SI as also suggested by Reviewer 2.

**Done. Now Fig. S34.**
* * *
Anonymous Referee #2

Do the authors give proper credit to related work and clearly indicate their own new/original contribution? *NO, literature on state-of-the-art is incomplete*

We will carefully update the cited references and pay due attention to the suggestions by the reviewer. Thanks for those hints.

**Done. L. 69 – 72.**

Is the overall presentation well structured and clear? *MOSTLY, some figures are not*

We will i) modify some of the figures according to the comments and suggestions by the reviewers (see also response to Reviewer 1).

**Done. See above.**

Should any parts of the paper (text, formulae, figures, tables) be clarified, reduced, combined, or eliminated? *Substantially clarify figures and/or reduce numbers of figures*

See previous comment. We will also move Fig. 4, 7, and 9 to the SI as was also suggested by Reviewer 1.

**Done. See above.**

Specific comments

Literature with respect to other related model work in this field is not sufficiently reviewed. References are sometimes unfortunate, e.g. MONERIS is cited not with original papers but with a side application and reference to GREAT-ER is outdated (see Kehrein et al., 2015). The EU approach (FOCUS model suite) is neglected as is the DRIPS model (Röpke et al., 2004).

We will carefully update the cited references and pay due attention to the suggestions by the reviewer. Thanks for those hints. Additional references include: (Behrendt, Kornmilch et al. 2002, Röpke, Bach et al. 2004, Kehrein, Berlekamp et al. 2015, Steffens, Jarvis et al. 2015, Villamizar and Brown 2017).

**We have included additional references as follows:**

**"Some of these models (e.g., SWAT (Arnold et al., 2011), MONERIS (Behrendt et al., 2002), GREAT-ER (Berlekamp et al., 2007;Kehrein et al., 2015)) or MACRO (Steffens et al., 2015;Larsbo et al., 2005) have been widely used, many others have been developed and used in specific research contexts (e.g, ZIN-AgriTra (Gassmann et al., 2013), SPIDER (Renaud et al., 2008;Villamizar and Brown, 2017) or DRIPS (Röpke et al., 2004))." L. 69 – 72.**

The assumption of general presence of persistent pesticides in baseflow and groundwater (page 6) is somehow terrifying to me, since prevention of elevated concentration levels in groundwater is the main driver for research in this field. Background concentrations determined in the calibration step need to be explicitly compared to the large number of data available. Fortunately, many of these data points are small and below critical limits. These data allow for defining an upper bound for the calibrated background concentrations.

The occurrence of (low) levels of background concentrations of some herbicides in groundwater is a well established fact (Vonberg, Vanderborght et al. 2014, Hakoun, Orban et al. 2017) and supported by our data sets (Gomides Freitas, Singer et al. 2008). An example for this persistence is provided with daily atrazine concentrations in river Rhine at Basel. Even years after the ban of this compound it is ubiquitously found.

[Figure]

Fig. 2: Temporal evolution of atrazine concentrations in river Rhine at Basel. Data source: IRMS.

The background concentration reflects the concentration in the groundwater which is an important contributor of water and substance during base flow conditions. The concentration in the groundwater is assumed to be proportional to the intensity of herbicide use in the area. Hence for simulations the calibrated background concentrations was proportional to areal fraction of the respective crop in a subcatchment. Thus a lower background concentration was introduced at subcatchments with a small share of agricultural area and vice versa a higher background concentration was selected for subcatchments with intensive agriculture land use (as mentioned in Table 1).

The means in the prior distributions of the background concentrations for the calibration were estimated from the values in Table S4.

**No changes were made.**

Many German cities situated in the Rhine catchment use combined sewer systems, whereby a significant fraction of urban surface runoff is directed not instantaneously into rivers (page 6), but to the nearest sewage treatment plant. While the substance-specific load reduction during wastewater treatment is included in the loss rate, it should be discussed whether retardation of transport into the receiving waters for the hydraulic retention time in the sewer system and the wastewater treatment plant could have an effect on the timing of the biocide concentration peaks in the simulations.

The short-term timing is less of an issue for the biocide simulations; the seasonal deviations are more striking. Additionally, the lag time for transport through the urban sewer system causes not a severe problem due to the daily time step used for our simulations. Nevertheless, the point is well taken and we will add a comment in the text.

**We added a sentence:**

**"The assumption of instantaneous transfer to the stream may cause some timing errors if compounds have residence times that are longer than the model time step (e.g. in wastewater treatment plants) but see the findings on routing effects in sec. 5.2. " L. 229 - 231.**

I think that some figures in the main text are not important enough to justify their presence. Moreover, the meaning of some figures is hardly recognizable from the discussion. I strongly encourage the authors to shift some of the figures to the supporting information. In the main text, I would focus the discussion on the most important points supported by meaningful and easily understandable figures with clearly labelled axes.

See previous comment. We will move Fig. 4, 7, and 9 to the SI as was also suggested by Reviewer 1.

**Done.**

- Figure 4: I could not find a description of the transformation step used to generate the data displayed here. Both axes are displayed in dimensionless scale, but I have no idea what that means. It remains also unclear, what the dashed line represents and why the Figure points to systematic deviations between observed and modelled concentrations (page 13, line 31).
  The reviewer is correct in that the figure was poorly labelled. As described on p. 10, L. 30, the calibration was performed on the Box-Cox-transformed data (Eq. 14, p, 10, L. 36 – p. 11, L: 2). The dashed line represents the regression of the residuals versus the observed concentration (in the transformed space). Without systematic deviations, the regression slope should not deviate from zero.
  We will add the necessary information in the figure and captions. Additionally, it will be moved to the SI. In the main text, we will better explain how to identify the systematic deviations.

**We moved the figure to the SI and improved its description.**

- Figure 5: It is not possible to extract the numbers for the GRI values given in the text (page 13/14) from the graph. For me, the range shown in the graph is larger with values up to 5 for the herbicides. Meaning of vertical lines is unclear and the error bars (grey) unreadable. Colour of second section (C-T, C-S) is not identifiable and it is confusing that all categories appear twice on the category axis.
  We will better label the figure and use additional colours to improve its readability. Furthermore, we make the link between statement in the text and the figure more implicit since it seems to be confusing.

**Done. Now Fig. 4.**

- Figure 10: Scaling of the axes is unclear (log-scale of what?)
  The axes represent the average density of data series in space (x-axis) and time (y-axis) across the model domain.

- The x axis represents the number of available data points X across the entire study area (e.g., how many discharge gauging stations, in log scale)
- The y axis gives the number of available data points Y per unit area per decade (, e.g., how many rainfall data for one location in space for a ten year period, in log scale)
- The radius of the circles is proportional to the total number of data points during a unit period of time for the entire study area (product X times Y, in log scale). We will modify the circles such that the circle area is proportional to the log(X Y).

**We have completely re-drawn this figure and hope that it is now easy to understand. Now Fig. 7.**

Model performance for biocides is poorer than for pesticides (page 14, line 2), which should be discussed in the light of the different number of calibrated parameters for herbicides (seven/nine with error function) and biocides (just one). Either the biocide model needs more detailed process descriptions or the herbicide model is over-parameterized mutually levelling off uncertainties introduced by the parameters.

The results suggest that the biocide model is overly simplistic because it cannot produce the observed seasonality. However, it is difficult to improve upon this aspect first because the underlying cause is not understood and second because of poor input data that preclude a more reliable simulation. We will add a sentence in the discussion to elaborate more on this issue beyond we already have on p. 17, L. 34 – 38.

**We added the sentence:**

**"Targeted surveys on actual use across the year might be a solution. Better input data could then allow to study further structural deficits of this very simple biocide model in more detail." L. 690 – 692.**

Minor comments

- pathways, wastewater (page 2)
  Will be corrected.

**Done. L. 12.**

- … where M(t) [g d-1] is the mass ratio … (page 5)
  We will reword to " is the rate of mass applied…"

**Done. L: 174.**

- … temperature sum models is used (page 5)
  Will be corrected.

**Done. L. 180.**

- … of the pronounced concentrations peaks (page 15)
  Will be corrected.

**Done. L. 604.**

- Number of Figure missing (page 16, line 2)
  The correct figure number will be inserted.

**Done. Fig. 6, L. 616.**

- … for the model compounds considered in this paper (page 16)
  Will be corrected.

**Done. L. 632.**

- …micropollutants from points sources (page 16)
  Will be corrected.

**Done. L. 640.**

- …purely hydrological models can accomplish (page 16)
  Will be corrected.

**Done. L. 656.**

- Number of Figure missing (page 17, last but one line)
  The correct figure number will be inserted.

**Done. L. 702.**

\*\*\*\*\*\*\*\*\*\*\*

Anonymous Referee #3

The authors present an extensive modeling study on pesticide transport in the Rhine basin. The paper including the supplement is very long (as is the river Rhine). It builds on quite an impressive amount of work and is well written, although not everything is explained in full detail. The latter is probably unavoidable with such kind of studies. I appreciate that the authors provide there input data and model code.

The basic idea is to set up a simple, parsimonious model, to calibrate it with detailed data (from Switzerland, where a number of very good, detailed studies were carried out in the past) and predict transport on large scale. The approach is actually quite successful. Modeled concentrations partly deviate from measured concentrations by up to a factor of 5 or so, but we should also not forget that the concentrations are low.

It is clear that the model can be improved when better input data become available (if ever).

I have only one major point to make. I refer to the last paragraph of section 5.1. The authors used a (lumped) modeling approach in which sorbed phase and solution phase concentration are expressed in total masses in the system (watershed). The model was parameterized with laboratory data from a study by Freitas et al. 2008 (reference missing), but it is not explained how.

Was the different soil to solution ratio in the lab considered? If the model is directly applied to the lab data (Ms=sorbed, Mw=dissolved), it would yield too low partitioning coefficients, what is indeed what the authors observed (section 5.1, last paragraph). This is so because, as a physicochemical fact, the sorption isotherm is independent of the soil to solution ratio as has been shown many times in the literature. In the lab, the mass in solution is much higher (by a factor of 5 or so, depending on the soil to solution ratio used). Taking it differently, sorption isotherms measured in the lab cannot be applied to a system (watershed) without making an assumption about the soil to solution ratio in it.

The reference was included in the reference list but as Gomides Freitas et al., 2008. We will correct this mistake and make the referencing consistent.

**Done. L. 445.**

The data stem from a field study (p. 11, L. 29 – 30). Pore water concentrations from soil samples were obtained upon extraction of 20 g of soil with 40 mL 0.01 M CaCl2 solution. Therefore, the reviewer is right in that the soil to solution ratio does not correspond to the conditions in field soils. How this fact affects the partitioning between the aqueous and the solid phase depends on the sorption isotherm. With a linear isotherm, the distribution coefficient Kd is constant irrespective of the soil-solution ratio. With non-linear isotherms this is not the case. Our field data indicate non-linear sorption behaviour that can described with a Freundlich isotherm.

Based on the experimental data, we can calculate how the (apparent) distribution coefficient Kd is expected to change upon a modification of the soil-solution ratio. To this end, we used the S-metolachlor data from (Gomides Freitas 2005) with a Freundlich exponent n of 0.7 and a Freundlich coefficient of 17 and calculated the distribution coefficient between solid and the aqueous phase for the lab conditions (see above, rho = 0.33, theta = 0.67) and for realistic field conditions (rho = 1.00, theta = 0.25). The resulting ratios between the lab and field distribution coefficients are depicted in Fig. 3 across a wide range of total concentrations (10 to 2500 microgram/kg soil).

These calculations first demonstrate that the Reviewer is correct in that the lab data underestimate sorption. Second, the data also reveal that the degree of underestimation is rather limited such that it won't affect our findings substantially. Nevertheless, we will add a comment stating that the different soil-solution ratios between the measuring conditions and the field situation will have contributed to the underestimation of partitioning.

**We added a sentence as follows:**

**"For sorption, this could be explained to some degree by different soil-water ratios of undisturbed soils and the conditions during the experimental procedure from which the priors were derived (Gomides Freitas, 2005). However, stronger sorption in the model could also compensate for pesticide applications that were missed by the model." L. 555 – 557.**

[Figure]

Fig. 3: Ratio between distribution coefficients in the field and in the lab for S-metolachlor based on the data reported in (Gomides Freitas 2005). The red line indicates identity between the lab and the field data.

Unfortunately, my print-out version did not contain page numbers.

page 1, line 17 to what degree

Will be corrected.

**Done. L. 17.**

At some locations in the text, references to figures lack the number (figure [empty]).

Will be corrected (see response above).

**Done.**

eq 1: M(t) is undefined (cf. eq 8)

This mistake will be corrected.

**Done. L. 167.**

eq 8: in eq 2 and 3, Ma was defined as the mass applied to the catchment the mass present in the catchment should be M as in the LHS of eq 1

Will be corrected.

**Done. L. 228.**

Figure 4: how were the residuals transformed?

All data are Box-Cox transformed, see response to Reviewer 1 on the same issue.

**Done.**

Figure 5 Overview of

Will be corrected.

**Done. Now Fig. 4.**

Figure 6: I did not get the rational behind the fold difference. This should be explained in the text and in the figure caption.

In the caption, we will refer to the description in Tab. 3 and we will add additional explanations in the text saying that this metric allows to quantify the deviation between observations and simulations irrespective of whether observations were under- or over estimated.

**We have inserted**

**"– indicating by which factor observations were over- or underestimated –" L. 585 – 586.**

Figure 7: What is the violin about? Is the envelope a standard error? Explain in the text and in the figure caption.

Violin plots are a combination of a boxplot and a kernel density plot. We will add a further explanation in text and captions and refer to the original publication introducing this kind of plot (Hintze and Nelson 1998).

**We have moved Fig. 7 to the SI and added "– combination of a boxplot and a kernel density plot (Hintze and Nelson, 1998) –" in the figure caption (now Fig. S33).**

Figure 9: Explain the envelope?

Violin plots are a combination of a boxplot and a kernel density plot. We will add a further explanation in text and captions and refer to the original publication introducing this kind of plot (Hintze and Nelson 1998).

**We have moved Fig. 9 to the SI and added "– combination of a boxplot and a kernel density plot (Hintze and Nelson, 1998) –" in the figure caption (now Fig. S34).**

Figure 10: I have no clue what this figure is about. Explain the axes, assign units. Explain properly or delete the whole thing.

We admit that the figure was poorly explained. We will add the information provided in the corresponding response to Reviewer 2 (see above).

**We have completely re-drawn this figure (now Fig. 7).**

Table 1: Did you properly explain the scaling factor ? I think no.

The scaling factor is introduced and explained on  p. 10, L. 20 – 26. In accordance with the reply to reviewer #1 on the same issue, we will add further explanation

**We added "The error scaling function makes the standard deviation of herbicide errors proportional to the remaining field stock to reflect that errors are larger in the application period than afterwards, when the compound is present in negligible amounts." L. 396 – 398.**

Table 2: Explain the abbreviations (like NADUF).

We will add these explanation (NADUF: NADUF – National long-term surveillance of Swiss rivers, IRMS: International Rhine Monitoring Station, NAWA SPEZ: National Surface Water Quality Monitoring Programme) and make sure they are used in consistent forms across the manuscript.

**We explained the abbreviations upon first mentioning and also added the explanations to the respective figure and table captions.**

Table 3: Explain the symbols in the caption.

We explain the symbols

**Done.**

Table 4: Explain abbreviations.

We will add these explanations.

**Done.**

Table 5: C and V should be defined. Atrazine

We will explain these symbols (C = calibration, V = validation). It is unclear what the reviewer wanted to say about atrazine.

**Done.**

***Summing up, the tables and figures should be self-explaining.

We agree and improve according to the responses above.

**Done.**

[revised manuscript text omitted]